# Sample Efficient Offline RL via T-Symmetry Enforced Latent State-Stitching

**Peng Cheng**[*‡1,4], **Zhihao Wu**[*1,4], **Jianxiong Li**[2], **Ziteng He**[2], **Haoran Xu**[3]
**Wei Sun**[2], **Youfang Lin**[1,4], **Xianyuan Zhan**[†2]
[1] Beijing Jiaotong University, [2] Tsinghua University, [3] University of Texas at Austin,
[4] Beijing Key Laboratory of Traffic Data Mining and Embodied Intelligence
`pcheng6@126.com`, `{zhwu, yflin}@bjtu.edu.cn`,
`zhanxianyuan@air.tsinghua.edu.cn`

## Abstract

Offline reinforcement learning (RL) has achieved notable progress in recent years. However, most existing offline RL methods require a large amount of training data to achieve reasonable performance and offer limited out-of-distribution (OOD) generalization capability due to conservative data-related regularizations. This seriously hinders the usability of offline RL in solving many real-world applications, where the available data are often limited. In this study, we introduce TELS, a highly sample-efficient offline RL algorithm that enables state-stitching in a compact latent space regulated by the fundamental time-reversal symmetry (T-symmetry) of dynamical systems. Specifically, we introduce a T-symmetry enforced inverse dynamics model (TS-IDM) to derive well-regulated latent state representations that greatly facilitate OOD generalization. A guide-policy can then be learned entirely in the latent space to optimize for the reward-maximizing next state, bypassing the conservative action-level behavioral regularization adopted in most offline RL methods. Finally, the optimized action can be extracted using the learned TS-IDM, together with the optimized latent next state from the guide-policy. We conducted comprehensive experiments on both the D4RL benchmark tasks and a real-world industrial control test environment, TELS achieves superior sample efficiency and OOD generalization performance, significantly outperforming existing offline RL methods in a wide range of challenging small-sample tasks.

## 1 Introduction

Offline reinforcement learning (RL) has seen rapid progress in recent years. It directly utilizes pre-collected datasets for policy learning, making them ideal for many real-world tasks that lack high-fidelity simulators or have restrictions on environment interaction (Levine et al., 2020; Zhan et al., 2022a). However, offline RL is also known to be prone to value overestimation, caused by extrapolation error when evaluating out-of-distribution (OOD) samples and amplified through the bootstrapped update procedure in RL (Kumar et al., 2019; Fujimoto et al., 2019). In the past few years, quite a few offline RL methods have been proposed, which commonly adopt the pessimism principle using strategies such as adding explicit or implicit policy constraints to prevent the selection of OOD actions (Kumar et al., 2019; Wu et al., 2019; Fujimoto et al., 2019; Fujimoto & Gu, 2021; Li et al., 2022), penalizing value function on unseen samples (Kumar et al., 2020; Bai et al., 2021; Lyu et al., 2022), or adopting in-sample learning to implicit regularize policy optimization (Kostrikov et al., 2022; Xu et al., 2023; Mao et al., 2024b; Wang et al., 2024; Zheng et al., 2024). Adopting such action-level constraints, although helpful to stabilize offline value and policy learning, also leads to over-conservatism and crippled OOD generalization performance (Li et al., 2022; Cheng et al., 2023). Most of the existing offline RL methods only perform well when trained with sufficiently large offline datasets with reasonable state-action space coverage (e.g., 1 million samples for simple

---

[*]Equal contribution.
[†]Corresponding Author.
[‡]Work done during internship at Institute for AI Industry Research (AIR), Tsinghua University.

D4RL benchmark tasks (Fu et al., 2020)). This forms a stark contrast to the reality in most real-world scenarios, such as industrial control (Zhan et al., 2022a; 2025), robotics (Sinha et al., 2022), and healthcare (Tang et al., 2022), where the real-world data are often scarce, and scaling up data collection can be rather costly.

Enhancing sample efficiency and OOD generalization capability is essential to making offline RL widely applicable to real-world applications. This is particularly important for small dataset settings, as most of the state-action space will become OOD regions. Several recent attempts have been made to improve the generalization performance of offline RL, which mainly follow three directions. The first direction builds upon the empirical observation that deep value functions interpolate well but struggle to extrapolate, thus allowing exploitation on interpolated OOD actions to promote generalization (Li et al., 2022). However, this method has a smoothness assumption on the offline dataset geometry and only applies to the continuous action space. The second class of methods avoids the conservative action-level constraint and instead performs reward maximization on the state-space (Xu et al., 2022a; Park et al., 2024), which allows exploitation of OOD actions as long as the corresponding state transitions are reachable (also referred to as "*state-stitching*"). Although these methods offer some promising generalization capabilities, they still require the state-action space to have reasonable data coverage to enable valid state-stitching. Finally, the last direction is to learn compact and robust latent representations to enhance sample efficiency. Most methods in this direction focus on extracting statistical-level information from the data, using techniques such as contrastive learning (Laskin et al., 2020; Agarwal et al., 2021a; Yang & Nachum, 2021; Uehara et al., 2021). Due to insufficient consideration of the underlying dynamics inside sequential data, these methods still struggle to provide generalizable information beyond data distribution. Some recent methods propose to learn representations that extract fundamental symmetries of dynamics to facilitate policy learning (Weissenbacher et al., 2022; Cheng et al., 2023), such as the time-reversal symmetry (T-symmetry) (Cheng et al., 2023), i.e., the underlying physical laws should not change under the time-reversal transformation. By leveraging such universally held symmetries in dynamical systems, it is possible to maximally promote OOD generalization without being restrained by data distribution-related information. Although promising, these methods are built upon offline RL backbone algorithms with action-level constraints (e.g., CQL (Kumar et al., 2020) or TD3+BC (Fujimoto & Gu, 2021)), which still suffer from the over-conservatism issue.

In this paper, we find that enabling state-stitching in a coherent, fundamental symmetry-enforced latent space can lead to a surprisingly strong sample-efficient offline RL algorithm. We refer to our method as Offline RL via **T**-symmetry **E**nforced **L**atent **S**tate-Stitching (TELS). Specifically, we introduce a T-symmetry enforced inverse dynamics model (TS-IDM) that can not only learn well-behaved and OOD generalizable latent state and action representations, but also facilitate effective action inference. Within the learned well-behaved latent state space, we can optimize a T-symmetry regularized guide-policy to output the next latent state that maximizes the accumulated reward, bypassing the conservative action-level behavioral regularization adopted in most offline RL algorithms. Lastly, the optimized action can be easily extracted by plugging the output of the guide-policy as the goal state in the learned TS-IDM. We evaluate TELS on both the challenging reduced-size D4RL benchmark tasks and a real-world industrial control test environment. Through comprehensive experiments, we show that TELS achieves state-of-the-art (SOTA) sample efficiency and OOD generalization capability, significantly outperforming existing offline RL algorithms on small datasets.

## 2 PRELIMINARIES

**Offline RL.** We consider the standard Markov decision process (MDP) setting (Sutton & Barto, 2018), which is represented as a tuple $\mathcal{M} = \{\mathcal{S}, \mathcal{A}, r, \mathcal{P}, \rho, \gamma\}$, and a dataset $\mathcal{D}$, which consists of trajectories $\tau = \{s_0, a_0, s_1, a_1, ..., s_T, a_T\}$. Here $\mathcal{S}$ and $\mathcal{A}$ denote the state and action spaces, $r(s, a)$ is a scalar reward function, $\mathcal{P}(s'|s, a)$ and $\rho$ denote the transition dynamics and initial state distribution respectively, and $\gamma \in (0, 1)$ is a discount factor. Our goal is to learn a policy $\pi(a|s)$ based on dataset $\mathcal{D}$ by maximizing the expected return in the MDP: $\mathbb{E}_\pi[\sum_{t=0}^\infty \gamma^t \cdot r(s_t, a_t)]$.

**Offline policy optimization in the state space.** Instead of adopting conservative action-level constraints for offline policy learning, Policy-guided Offline RL (POR) (Xu et al., 2022a) proposes to decompose the conventional reward-maximizing policy into a guide-policy and an execute policy.

The guide-policy only works in the state space to find the optimal next state that maximizes the state-value function, and the execute-policy is learned as an inverse dynamics model (Xu et al., 2022a) or a goal-conditioned imitative policy (Park et al., 2024). Such methods only need to learn a state-only value function $V$ using the IQL-style expectile regression, as proposed by Kostrikov et al. (2022), or the sparse value learning objective as discussed in (Xu et al., 2023). We present the former as follows:

$$V = \arg\min_{V} \mathbb{E}_{(s,r,s') \sim \mathcal{D}} \left[ L_2^{\tau} \left( r(s) + \gamma \bar{V}(s') - V(s) \right) \right]. \tag{1}$$

where $L_2^{\tau}(x) = |\tau - \mathbb{1}(x < 0)|x^2$ is the asymmetric expectile regression loss and $\bar{V}$ denotes the target value network. Based on the learned state-value function, we can learn a guide-policy $\pi_g(s'|s)$ to serve as a prophet by telling which state the agent should (high reward) and can (logical generalization) go to, without being constrained to state-action transitions seen in the dataset. This can be achieved by leveraging an advantage weighted regression (AWR) objective (Neumann & Peters, 2008; Peng et al., 2019) to maximize the value while implicitly constraining $\pi_g$ to $s \to s'$ transitions observed in the dataset (i.e., *state-stitching*):

$$\pi_g = \arg\max_{\pi_g} \mathbb{E}_{(s,r,s') \sim \mathcal{D}} \left[ \exp(\alpha \cdot A(s,s')) \log \pi_g(s' \mid s) \right]. \tag{2}$$

where the advantage $A(s,s') = r + \gamma V(s') - V(s)$ serves as the behavior cloning weight, and $\alpha$ is the temperature parameter to prioritize value maximization over state-wise imitation.

For the execute-policy $\pi_e$, POR employs a supervised learning framework and trains $\pi_e$ by maximizing the likelihood of the actions given the states and next states: $\max_{\pi_e} \mathbb{E}_{(s,a,s') \sim \mathcal{D}}[\log \pi_e(a \mid s, s')]$. During evaluation phase, given the current state $s$, we can sample the optimized next state $s'$ from $\pi_g(s'|s)$, and get final action simply as $a^* = \pi_e(a \mid s, \pi_g(s'|s))$.

**Time-reversal symmetry for generalizable offline RL.** Recently, leveraging fundamental, universally held symmetries of dynamics like T-symmetry discovered in classical and quantum mechanics (Lamb & Roberts, 1998; Huh et al., 2020) has been shown to be a promising approach to enhance the generalization of offline RL (Cheng et al., 2023; Zhan et al., 2025). Specifically, if we model the system dynamics with measurements $\mathbf{x}$ as a set of non-linear first-order differential equations (ODEs) expressed as $\frac{d\mathbf{x}}{dt} = F(\mathbf{x})$, a dynamical system is said to exhibt *time-reversal symmetry* if there is an invertible transformation $\Gamma$ that reverses the direction of time: i.e., $d\Gamma(\mathbf{x})/dt = -F(\Gamma(\mathbf{x}))$. For the discrete-time MDP setting, the T-symmetry can be extended as learning a pair of ODE forward dynamics $F(s,a) \to \dot{s}$ and reverse dynamics $G(s',a) \to -\dot{s}$, and require them to satisfy $F(s,a) = -G(s',a)$, where the time-derivative of state $\dot{s} = \frac{ds}{dt}$ is approximated as $s' - s$.

Based on this intuition, TSRL (Cheng et al., 2023) constructed an encoder-decoder structured *T-symmetry enforced dynamics model* (TDM) for representation learning, which embeds a pair of latent ODE forward and reverse dynamics to enforce T-symmetry. TSRL achieves impressive performance under small-sample settings, and its variant has been successfully deployed for real-world industrial control (Zhan et al., 2025), but it still has some limitations. First, TSRL only uses the learned encoder from TDM to derive the latent representations, without fully exploiting the rich dynamics-related information for downstream policy learning. Second, its representation learning scheme uses both state and action as inputs, forcing TSRL to involve policy-induced actions during policy optimization, which inevitably requires adding a conservative action-level behavioral constraint as in TD3+BC (Fujimoto & Gu, 2021) to stabilize training. Moreover, involving action as an input for representation learning is also prone to capturing biased behaviors in the behavioral policy, which could impede learning fundamental, distribution-agnostic dynamics patterns in data. Please refer to Appendix A for a more detailed comparison and discussion.

## 3 OFFLINE RL VIA T-SYMMETRY ENFORCED LATENT STATE-STITCHING

We now present our proposed method, TELS, which comprises a T-symmetry enforced inverse dynamics model (TS-IDM) integrated with an effective offline policy optimization procedure operated in latent state space (illustrated in Figure 1).

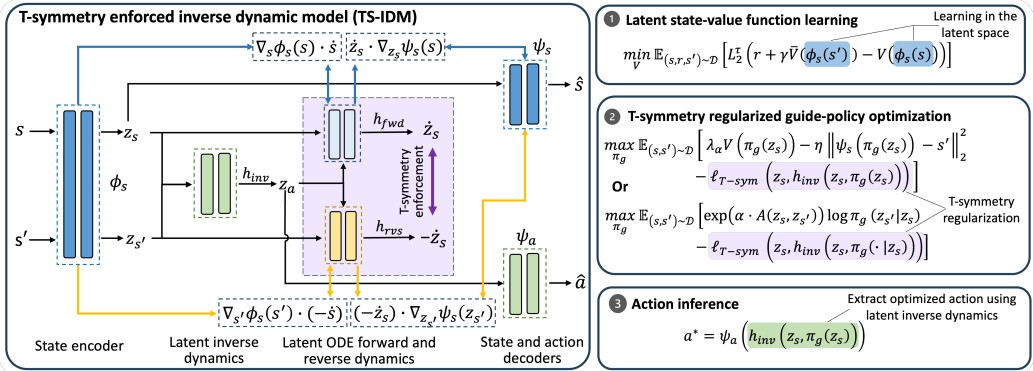

Figure 1: Overview of **T**-symmetry **E**nforced **L**atent **S**tate-Stitching (TELS) framework. **Left:** The illustration of TS-IDM structure. **Right:** The process of training T-symmetry regularized guided-policy.

## 3.1 T-SYMMETRY ENFORCED INVERSE DYNAMIC MODEL

As illustrated in Figure 1, if inspecting the input and output of our proposed TS-IDM, it functions similarly to an inverse dynamics model that takes current and next state $(s, s')$ as input and outputs the predicted action $a$. However, in its interior, TS-IDM comprises a state encoder $\phi_s(s) = z_s$ and a corresponding decoder $\psi_s(z_s) = \hat{s}$; a latent inverse dynamics module $h_{inv}(z_s, z_{s'}) = z_a$ followed by an action decoder $\psi_a(z_a) = \hat{a}$; and most importantly, a pair of T-symmetry enforced latent ODE forward and reverse dynamics predictors $h_{fwd}(z_s, z_a) = \dot{z}_s$ and $h_{rvs}(z_{s'}, z_a) = -\dot{z}_s$. All these sub-components are implemented as simple 2-layer MLPs. In the following, we will dive into the design intuitions and learning objectives of these components.

**Encoding and decoding.** As previously discussed, constructing an informative and well-structured latent space is critical for sample-efficient offline policy optimization. To this end, we introduce a state encoder $\phi_s(s) = z_s$ to map a state $s$ into corresponding latent representation $z_s$, and also a state decoder $\psi_s(z_s) = s$ to reconstruct the original state from its latent embedding, ensuring that the learned latent representations remain faithful to the original state space and avoid excessive distortion.

We then construct a latent inverse dynamics module $h_{inv}(z_s, z_{s'}) = z_a$, which infers the latent action $z_a$ from the latent state transitions $(z_s, z_{s'})$. By inferring actions from state transitions, the learned latent space implicitly encodes the underlying dynamics of the environment. Moreover, the inverse dynamic module $h_{inv}$ can be integrated with a pair of latent ODE dynamics predictors ($h_{fwd}$ and $h_{rvs}$) to derive the T-symmetry property of the system, which we will introduce in more detail shortly. Finally, to ensure that the inferred actions are both meaningful and interpretable, we employ an action decoder $\psi_a(z_a) = \hat{a}$ to map the latent action back to its original action space. We can thus formulate the reconstruction loss for the states and actions as follows:

$$\ell_{\text{rec}}(s, a, s') = \underbrace{\|\psi_s(\phi_s(s)) - s\|_2^2}_{\text{reconstruction loss of states}} + \underbrace{\|\psi_a(h_{inv}(z_s, z_{s'})) - a\|_2^2}_{\text{reconstruction loss of actions}}. \quad (3)$$

**Latent ODE forward and reverse dynamics.** Drawing inspiration from previous research that integrates physics-informed insights into dynamical systems modeling (Brunton et al., 2016; Champion et al., 2019; Huh et al., 2020; Cheng et al., 2023), we embed a pair of latent ODE forward and reverse dynamics $h_{fwd}(z_s, z_a) = \dot{z}_s$ and $h_{rvs}(z_{s'}, z_a) = -\dot{z}_s$ to separately capture the forward and reverse time evolution in the latent states. We are interested in modeling ODE systems because it encourages learning parsimonious models helpful to uncover fundamental properties from the data that can maximally promote generalization (Brunton et al., 2016; Champion et al., 2019). Note that based on the chain rule, we can derive the supervision signal for the latent dynamics modules with $\dot{z}_s = \frac{dz}{dt} = \frac{dz_s}{ds} \cdot \frac{ds}{dt} = \nabla_s z_s \cdot \dot{s} = \nabla_s \phi_s(s) \cdot \dot{s}$ to enforce the ODE property. Therefore, we introduce the following training losses for $h_{fwd}$ and $h_{rvs}$:

$$\ell_{\text{dyn}}(s, s') = \underbrace{\|(\nabla_s z_s)\dot{s} - \dot{z}_s\|_2^2}_{\text{latent ODE forward dynamics}} + \underbrace{\|(\nabla_{s'} z_{s'})(-\dot{s}) - (-\dot{z}_s)\|_2^2}_{\text{latent ODE reverse dynamics}}$$

$$= \|\nabla_s \phi_s(s)\dot{s} - h_{fwd}(z_s, z_a)\|_2^2 + \|\nabla_{s'} \phi_s(s')(-\dot{s}) - h_{rvs}(z_{s'}, z_a)\|_2^2, \quad (4)$$

where the latent action $z_a$ is obtained from the latent inverse dynamics module $h_{inv}(z_s, z_{s'})$.

**ODE property enforcement on state decoder.** Note that in $\ell_{\mathrm{dyn}}(s, s')$, we actually implicitly enforced the ODE property on the state encoder $\phi_s$, the same should also apply to the state decoder $\psi_s$ to ensure compatibility with the T-symmetry formalism, i.e. the time-derivative of the state encoder $\frac{d\phi_s(s)}{dt}$ and decoder $\frac{d\psi_s(z_s)}{dt}$ should behave in the same way as $\dot{z}_s$ and $\dot{s}$. Similar to the previous treatment on the state encoder, as $\dot{s} = \frac{d\psi_s(z_s)}{dt} = \frac{d\psi_s(z_s)}{dz_s} \cdot \frac{dz_s}{dt} = \nabla_{z_s}\psi_s(z_s) \cdot \dot{z}_s$, we can use the following objective to enforce the ODE property for the state decoder $\psi_s$:

$$\ell_{\mathrm{ode}}(s, s') = \underbrace{\|\nabla_{z_s}\psi_s(z_s) \cdot \dot{z}_s - \dot{s}\|_2^2}_{\text{enforce ODE of } \psi_s \text{ on } h_{fwd}} + \underbrace{\|\nabla_{z_{s'}}\psi_s(z_{s'}) \cdot (-\dot{z}_s) - (-\dot{s})\|_2^2}_{\text{enforce ODE of } \psi_s \text{ on } h_{rvs}}$$

$$= \|\nabla_{z_s}\psi_s(z_s) \cdot h_{fwd}(z_s, z_a) - \dot{s}\|_2^2 + \|\nabla_{z_{s'}}\psi_s(z_{s'}) \cdot h_{rvs}(z_{s'}, z_a) + \dot{s}\|_2^2. \quad (5)$$

Notably, the ODE property enforcement in Eq. (5) is not considered in the T-symmetry enforced dynamics model (TDM) proposed by TSRL (Cheng et al., 2023). In other words, TDM only enforces the ODE properties for encoders but not for decoders. This can cause inconsistency between the learned dynamics and the underlying ODE structure, leading to inaccurate ODE representations.

**T-symmetry enforcement.** To further regularize the learned latent representations, we enforce T-symmetry by requiring $h_{fwd}(z_s, z_a) = -h_{rvs}(z_{s'}, z_a)$, which corresponds to the following loss:

$$\ell_{\mathrm{T-sym}}(z_s, z_a) = \|h_{fwd}(z_s, z_a) + h_{rvs}(z_s + h_{fwd}(z_s, z_a), z_a)\|_2^2. \quad (6)$$

where we use the fact that $z_{s'} = z_s + \dot{z}_s = z_s + h_{fwd}(z_s, z_a)$ and $h_{rvs}(z_s + h_{fwd}(z_s, z_a), z_a) = -\dot{z}_s = -h_{fwd}(z_s, z_a)$ to further couple the learning process of $h_{fwd}$ and $h_{rvs}$. Moreover, given a latent state-action pair $(z_s, z_a)$, the above T-symmetry consistency loss can also serve as an evaluation metric to assess their agreement with the learned TS-IDM. A large T-symmetry loss indicates that the latent state-action representation $(z_s, z_a)$ induced by some $(s, s')$ may not satisfy the fundamental dynamics pattern, making it more likely to be a problematic or non-generalizable sample.

**Overall learning objective.** Finally, the complete training loss function of TS-IDM is as follows:

$$\mathcal{L}_{\mathrm{TS-IDM}} = \sum_{(s,a,s')\in\mathcal{D}} \left[\ell_{\mathrm{rec}} + \beta \cdot (\ell_{\mathrm{dyn}} + \ell_{\mathrm{ode}} + \ell_{\mathrm{T-sym}})\right](s, a, s'). \quad (7)$$

where $\beta$ is a hyperparameter that balances extracting fundamental dynamics properties and ensuring the interpretability of the learned representation. Note that we employ a single shared $\beta$ for $\ell_{\mathrm{dyn}}$, $\ell_{\mathrm{ode}}$, and $\ell_{\mathrm{T-sym}}$ terms. This is to ensure that the ODE property and T-symmetry regularization are enforced on a consistent scale for these strongly coupled loss terms, while also reducing the number of unnecessary hyperparameters. This is actually critical, as we have empirically shown in Appendix B.5, using a shared $\beta$ enables stable training, while adopting separate weights can cause a substantial performance drop. Despite containing multiple sub-modules, our proposed TS-IDM is actually quite small (based on several simple MLP layers), which can be efficiently and stably learned owing to its highly coupled design. The entire training process can be completed in merely 20 minutes and 5 minutes in our PyTorch and JAX implementations, respectively (see Table 12 in Appendix).

## 3.2 LATENT SPACE OFFLINE POLICY OPTIMIZATION

Once we have learned TS-IDM, we can extract three highly useful components from it to facilitate sample-efficient downstream offline policy optimization, including: 1) the state encoder $\phi(s)$ that provides an ideal, well-behaved latent space for state-stitching; 2) T-symmetry consistency as an additional regularizer to prevent erroneous generalization when learning a guide-policy in the latent state space; and 3) the TS-IDM itself can serve as an execute-policy as in POR (Xu et al., 2022a) to extract optimized action given a learned guide-policy.

**Latent state-value functions learning.** Based on the state encoder $\phi_s(s)$ from the learned TS-IDM, we can convert the entire offline policy optimization process into the latent state space, which enjoys both a stable learning process and generalizability due to more compact and well-behaved representations. Specifically, we can use a similar expectile regression loss as in Eq. (1) to learn a state-value function $V(z_s)$, but in the latent state space:

$$\min_V \mathbb{E}_{(s,r,s')\sim\mathcal{D}}\left[L_2^\tau\left(r + \gamma\bar{V}\left(\phi_s(s')\right) - V\left(\phi_s(s)\right)\right)\right]. \quad (8)$$

**T-symmetry regularized guide-policy optimization.** A key benefit of learning within the T-symmetry preserving latent space is that, as T-symmetry captures what is essential and invariant about the dynamical system, it can provide generalizable information even for OOD samples beyond the offline dataset. This naturally favors learning a reward-maximizing guide-policy $\pi_g$ in the latent space, which can enjoy more effective state-stitching. Moreover, by leveraging the T-symmetry consistency term $\ell_{\text{T-sym}}(\cdot)$ in Eq. (6) as an additional regularizer, we can prevent $\pi_g$ from outputting problematic and non-generalizable latent next state, thereby further enhancing logical state-wise OOD generalization. In TELS, we provide two instantiations for guide-policy optimization, depending on the choice of using deterministic policy $\pi_g(z_s)$ or stochastic policy $\pi_g(z_{s'}|z_s)$:

- **Deterministic policy:**

$$\max_{\pi_g} \mathbb{E}_{(s,s')\sim\mathcal{D}}\Big[\lambda_\alpha V(\pi_g(z_s)) - \eta\|\psi_s(\pi_g(z_s)) - s'\|_2^2 - \ell_{\text{T-sym}}\left(z_s, h_{inv}\left(z_s, \pi_g(z_s)\right)\right)\Big] \quad (9)$$

- **Stochastic policy:**

$$\max_{\pi_g} \mathbb{E}_{(s,s')\sim\mathcal{D}}\Big[\exp(\alpha \cdot A(z_s, z_{s'})) \log \pi_g(z_{s'} \mid z_s) - \ell_{\text{T-sym}}(z_s, h_{inv}(z_s, \pi_g(\cdot|z_s)))\Big] \quad (10)$$

where $z_s = \phi_s(s)$, $z_{s'} = \phi_s(s')$, and $A(z_s, z_{s'}) = r + \gamma V(z_{s'}) - V(z_s)$. For the deterministic policy $\pi_g(z_s)$, we optimize the guide-policy by maximizing the latent state-value function $V$ weighted by a normalization term $\lambda_\alpha$, together with two extra regularization terms. The first regularizes the next state decoded from the guide-policy using state decoder $\psi_s$ should not deviate too much from the next state $s'$ in the dataset. The second term regularizes the guide-policy induced latent state-action pair (i.e., $(z_s, z_a) = (z_s, h_{inv}(z_s, \pi_g(z_s)))$) to comply with the T-symmetry consistency specified in the learned TS-IDM. For the stochastic guide-policy $\pi_g(z_{s'}|z_s)$, we adopt a similar AWR-style objective as in Eq. (2), while also incorporating the T-symmetry consistency regularization as in the deterministic version. In our experiments, we find that the deterministic version of the objective Eq. (9) works well for the MuJoCo locomotion tasks, while the stochastic version Eq. (10) works better for more complex Antmaze tasks, potentially due to the more stochastic nature of the task environment.

**Action inference.** After learning the guide-policy $\pi_g$, we can further use it to extract the optimized action for control. To do this, we can simply use the optimized latent next state $z_{s'}^*$ obtained from guide-policy $\pi_g(z_s)$ or $\pi_g(\cdot|z_s)$ as the goal state, and plug it into the learned latent inverse dynamics module $h_{inv}(z_s, z_{s'})$ in TS-IDM to replace $z_{s'}$. The final action can be extracted by decoding the resulting latent action from $h_{inv}$ using the action decoder $\psi_a$ :

$$a^* = \psi_a\left(h_{inv}\left(z_s, \pi_g(z_s)\right)\right). \quad (11)$$

Note that there is no training process needed for this stage. Moreover, throughout our policy optimization process, actions are not involved, allowing TELS to completely bypass the conservatism issue caused by the action-level regularization. Please refer to Algorithm 1 in Appendix C for the detailed implementation, as well as the training and inference procedure of TELS.

## 4 EXPERIMENTS

In this section, we present the evaluation results of TELS on the D4RL benchmark tasks (Fu et al., 2020) against behavior cloning (BC), and existing offline RL methods: TD3+BC (Fujimoto & Gu, 2021), CQL (Kumar et al., 2020), IQL (Kostrikov et al., 2021), DOGE (Li et al., 2022), POR (Xu et al., 2022a), model-based methods MOPO (Yu et al., 2020) and COMBO (Yu et al., 2021b), diffusion-based method IDQL (Hansen-Estruch et al., 2023), and TSRL (Cheng et al., 2023), the current SOTA method in small-sample settings. To demonstrate the effectiveness of TELS in solving real-world tasks, we also validate TELS in a real-world industrial control environment, which is a data center (DC) cooling control testbed built by a recent work (Zhan et al., 2025). Moreover, we conduct additional experiments to evaluate the OOD generalizability of TELS on a challenging task, and the strengths of the representations learned with TS-IDM in improving small-sample performance.

Table 1: Normalized scores on reduced-size D4RL datasets (averaged over the final 10 evaluations with 5 seeds). We report the standard deviations after the $\pm$ sign. Numbers at or above 95% of the best value in the row are highlighted in **bold**.

| Task | Size (ratio) | BC | TD3+BC | MOPO | COMBO | CQL | IQL | DOGE | IDQL | POR | TSRL | TELS |
|---|---|---|---|---|---|---|---|---|---|---|---|---|
| Hopper-m | 10k (1%) | 29.7±11.7 | 40.1±18.6 | 5.5 ± 2.3 | 30.2 ± 28.0 | 43.1±24.6 | 46.7±6.5 | 44.2 ± 10.2 | 44.2±12.1 | 46.4 ± 1.7 | 62.0±3.7 | **77.3 ± 10.7** |
| Hopper-mr | 10k (2.5%) | 12.1±5.3 | 7.3±6.1 | 6.8 ± 0.3 | 10.6 ± 13.1 | 2.3±1.9 | 13.4±3.1 | 17.9 ± 4.5 | 21.7±7.0 | 17.4 ± 6.2 | 21.8±8.2 | **43.2 ± 3.5** |
| Hopper-me | 10k (0.5%) | 27.8±10.7 | 17.8±7.9 | 5.8 ± 5.8 | 13.9 ± 22.0 | 29.9±4.5 | 34.3±8.7 | 50.5 ± 25.2 | 43.2±4.4 | 37.9 ± 6.1 | 50.9±8.6 | **100.9 ± 6.8** |
| Halfcheetah-m | 10k (1%) | 26.4±7.3 | 16.4±10.2 | -1.1 ± 4.1 | 16.5 ± 2.4 | 35.8±3.8 | 29.9±0.12 | 36.2 ± 3.4 | 36.4±1.5 | 33.3±3.2 | 38.4±3.1 | **40.8 ± 0.6** |
| Halfcheetah-mr | 10k (5%) | 14.3±7.8 | 17.9±9.5 | 11.7 ± 5.2 | 11.8 ± 15.3 | 8.1±9.4 | 22.7±6.4 | 23.4 ± 3.6 | 26.7±1.0 | 27.5±3.6 | 28.1±3.5 | **33.2 ± 1.0** |
| Halfcheetah-me | 10k (0.5%) | 19.1±9.4 | 15.4±10.7 | -1.1 ± 1.4 | 5.2 ± 6.1 | 26.5±10.8 | 10.5±8.8 | 26.7 ± 6.6 | 38.8±1.9 | 34.7±2.6 | **39.9 ± 21.1** | **40.7 ± 1.2** |
| Walker2d-m | 10k (1%) | 15.8±14.1 | 7.4±13.1 | 3.1 ± 4.7 | 3.6 ± 1.1 | 18.8±18.8 | 22.5±3.8 | 45.1 ± 10.2 | 31.7±14.2 | 22.2±3.6 | 49.7±10.6 | **62.4 ± 5.3** |
| Walker2d-mr | 10k (3.3%) | 1.4±1.9 | 5.7±5.8 | 3.3 ± 2.7 | 4.2 ± 15.6 | 8.5±2.19 | 10.7±11.9 | 13.5 ± 8.4 | 12.2±10.5 | 14.8±4.2 | 26.0±11.3 | **54.8 ± 6.0** |
| Walker2d-me | 10k (0.5%) | 21.7±8.2 | 7.9±9.1 | 0.6 ± 2.7 | 0.1 ± 0.1 | 19.1±14.4 | 26.5±8.6 | 35.3 ± 11.6 | 21.8±14.5 | 20.1±8.6 | 46.4±17.4 | **87.4 ± 13.3** |
| Antmaze-u | 10k (1%) | 44.7 ± 42.1 | 0.7 ± 1.2 | 0.0 | 0.0 | 5.5 ± 2.3 | 65.1 ± 19.4 | 56.3 ± 24.4 | 67.5 ±12.4 | 6.1 ± 7.3 | 76.1 ± 15.6 | **88.7 ± 7.7** |
| Antmaze-u-d | 10k (1%) | 24.1 ± 22.2 | 16.27 ± 16.4 | 0.0 | 0.0 | 0.5 ± 0.1 | 34.6 ± 18.5 | 41.7 ± 18.9 | 55.1 ± 36.8 | 42.1 ± 14.2 | 52.2 ± 22.1 | **60.9 ± 16.9** |
| Antmaze-m-d | 100k (10%) | 0.0 | 0.0 | 0.0 | 0.0 | 0.0 | 4.8 ± 5.9 | 0.0 | 9.0 ±3.4 | 0.0 | 0.0 | **47.2 ± 17.3** |
| Antmaze-m-p | 100k (10%) | 0.0 | 0.0 | 0.0 | 0.0 | 0.0 | 12.5 ± 5.4 | 0.0 | 9.4 ± 14.7 | 0.0 | 0.0 | **62.9 ± 17.8** |
| Antmaze-l-d | 100k (10%) | 0.0 | 0.0 | 0.0 | 0.0 | 0.0 | 3.6 ± 4.1 | 0.0 | 16.1 ± 8.4 | 0.0 | 0.0 | **39.8 ± 14.1** |
| Antmaze-l-p | 100k (10%) | 0.0 | 0.0 | 0.0 | 0.0 | 0.0 | 3.5 ± 4.1 | 0.0 | 9.7 ±8.5 | 0.0 | 0.0 | **47.3 ± 13.1** |
| Pen-human | 5k (100%) | 34.4 | 8.4 | 9.7 | 27.7 | 37.5 | 71.5 | 42.6 ± 16.3 | 67.9 ± 17.3 | 64.1 ± 25.3 | **80.1 ± 18.1** | **77.4 ± 17.2** |
| Hammer-human | 5k (100%) | 1.5 | 2.0 | 0.2 | 0.2 | **4.4** | 1.4 | -1.2 ± 0.2 | 2.7 ± 1.3 | 0.2 ± 0.1 | 0.2 ± 0.3 | 3.6 ± 1.5 |
| Door-human | 5k (100%) | 0.5 | 0.5 | -0.2 | -0.3 | 9.9 | 4.3 | -1.1 ± 0.2 | 10.5 ± 1.5 | 0.1 ± 0.1 | 0.5 ± 0.3 | **11.8 ± 1.6** |
| Relocate-human | 5k (100%) | 0.0 | -0.3 | -0.2 | -0.3 | 0.2 | 0.1 | 0.1 ± 0.2 | 0.2 ± 0.1 | 0.1 ± 0.1 | 0.1 ± 0.1 | 0.3 ± 0.2 |

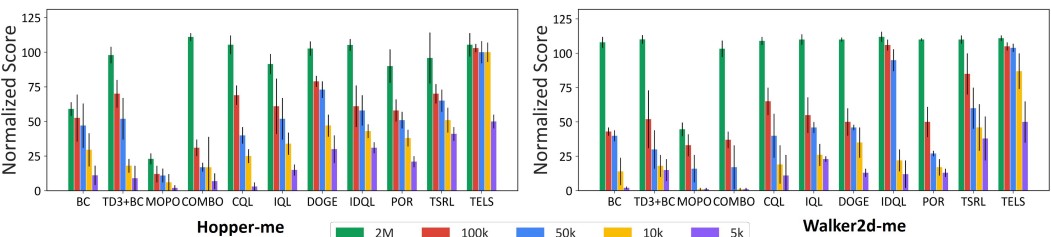

Figure 2: Performance of TELS against baselines under different data sizes. The error bars represent the standard deviation calculated over 5 random seeds.

## 4.1 COMPARATIVE EVALUATION ON SMALL-SAMPLE SETTING

**Evaluation on D4RL benchmarks.** In Table 1, we evaluate TELS against baseline methods on challenging reduced-size D4RL datasets (5k∼100k samples, about 0.5∼10% of their original sizes)*. These small-sample tasks are particularly challenging for offline RL algorithms, as the data only sparsely cover the state-action space and require strong OOD generalization capability for algorithms to achieve reasonable performance. Results on full D4RL datasets can be found in Appendix B.1.

As shown in Table 1, most baselines fail to learn reasonable policies under small datasets, especially in the challenging 100k Antmaze-medium/large datasets. For example, conventional offline RL methods like TD3+BC and CQL perform poorly on small datasets, primarily due to their over-conservative data-related policy constraints. Model-based methods also perform badly due to insufficient samples to learn accurate dynamics models and the use of problematic model rollout data. Baselines that have generalization promotion designs, such as DOGE and TSRL, perform slightly better but still fail miserably in the challenging Antmaze-m/l tasks, as they still adopt conservative action-level constraints to stabilize policy learning. Recent diffusion-based methods like IDQL, although perform well on large datasets, struggle to learn when given limited data. By contrast, TELS dominates the chart and outperforms all other baselines in all tasks, sometimes by a large margin. This is attributed to the leverage of fundamental, data distribution-agnostic T-symmetry property for policy learning, which greatly improves the OOD generalization performance. This is evident when observing the huge performance difference between POR and TELS, as the former shares a similar policy optimization procedure but does not use the T-symmetry enforced representation and policy regularization.

We also evaluate the performance of the algorithms across different dataset sizes in Figure 2. The results show that TELS can robustly maintain reasonable performance even with only 5k samples,

---

*We use the same reduced-size MuJoCo datasets from the TSRL paper (Cheng et al., 2023), and randomly sub-sample 100k Antmaze datasets for experiments. We use the original Adroit-human datasets for evaluation, as they are already small.

Table 2: Evaluation results in the real-world DC cooling control testbed (6-hour length experiments). Results with the lowest ACLF under zero thermal safety violations are highlighted in bold.

| Testbed | CQL | IQL | TSRL | TELS |
|---|---|---|---|---|
| Server energy consumption (kWh) | 41.44 | 39.80 | 40.30 | 40.61 |
| ACU energy consumption (kWh) | 4.16 | 16.27 | 10.95 | 8.19 |
| Energy efficiency measure: ACLF (the lower the better) | 10.3% | 40.89% | 27.16% | **20.17%** ↓ |
| Percentage of thermal safety violation (the lower the better) | 40.99% | 0.00% | 0.00% | **0.00**% |

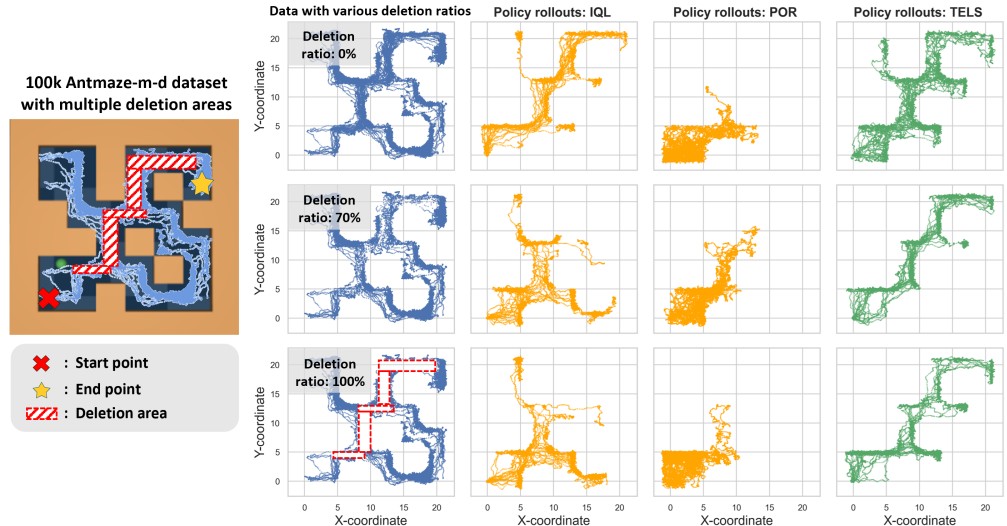

Figure 3: **Left:** Illustration of the 100k Antmaze-m-d task with multiple deletion areas, where the red cross denotes the start point, the yellow star denotes the goal locations, and the red shaded areas denote the data deletion regions. **Right:** Visualization of the training dataset and policy rollout trajectories generated by trained policies from various algorithms under varying deletion ratios.

surpassing all the other methods, while most baseline methods suffer from a significant performance drop when training samples are decreased.

**Evaluation on real-world industrial control test environment.** To further demonstrate the effectiveness of TELS in solving real-world industrial control tasks, we deploy TELS in a real-world DC cooling control testbed (Zhan et al., 2025) and compare against CQL, IQL, and TSRL. The testbed comprises 22 servers with oscillating server loads and an Air-Cooling Unit (ACU) for cooling control. A small historical operational dataset (43k real-world samples collected over 61 days) with 105 state-action features is used for policy learning. The goal is to improve the energy efficiency of the DC's cooling systems (minimizing the Air-side Cooling Load Factor (ACLF), calculated as the ratio of energy consumption of ACU to servers), while satisfying thermal safety constraints (no overheating). We follow the same real-world experiment setup as in (Zhan et al., 2025) and present the details in Appendix D.2. As shown in Table 2, under a similar server energy consumption level, TELS learns the best control policy, achieving 20.17% ACLF while maintaining zero thermal safety violations. CQL learns a naive policy that achieves lower ACLF but with significant thermal safety violations. This shows TELS's effectiveness in solving real-world complex industry control tasks.

**OOD generalization capability.** To further examine the OOD generalizability of TELS, we construct a very challenging task based on the reduced-size 100k Antmaze-m-d dataset, as illustrated in Figure 3. Specifically, we randomly remove samples within 5 critical regions along the critical paths from the start to the goal locations. This task requires extremely strong OOD generalization capability to solve, as the vital information for the optimal trajectory is extremely scarce or completely OOD. We train IQL, POR, and TELS on the remaining data and plot their policy rollouts over 20 episodes for performance evaluation and behavior analyses (due to page limit, we include results for IDQL, DOGE, TSRL in Appendix B.3). As shown in Figure 3, IQL can only achieve some success when the deletion ratio is 0%, and POR fails to reach the goal in all cases. By contrast,

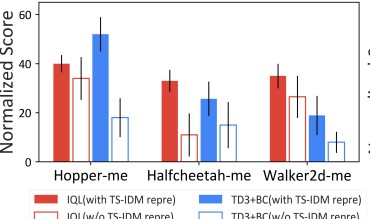 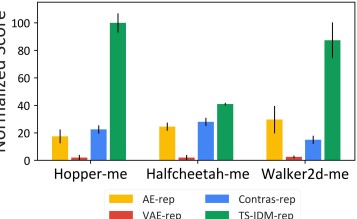 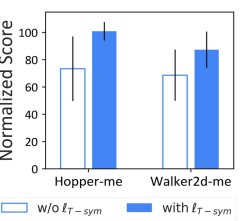

Figure 4: **Left:** Performance of IQL and TD3+BC on 10k datasets with or without using the representation from TS-IDM. **Right:** The performance of TELS with different representation models on 10k datasets. The error bars represent the standard deviation calculated over 5 random seeds.

Figure 5: Impact of $\ell_{\text{T-sym}}$ on policy optimization.

TELS consistently learns an optimal policy even with 70% and 100% deletion rates. It can effectively utilize the limited information provided in the sparse remaining data samples at the boundaries of the deletion areas for policy learning. These highlight the extraordinary OOD generalization capability of TELS in extremely challenging low-data regimes.

## 4.2 ANALYSIS AND ABLATION OF TELS

**Ablations on the design of TS-IDM.** To examine the impact of each sub-module in TS-IDM, we evaluate various variants of TS-IDM, starting with a vanilla latent inverse dynamics module with encoder and decoders, denoted as " $\phi/\psi + h_{inv}$ ", gradually adding latent forward and reverse dynamics " $h_{fwd}, h_{rvs}$ ", ODE property enforcement " $\ell_{\text{ode}}$ ", and eventually the T-symmetry consistency loss " $\ell_{\text{T-sym}}$ ", resulting in the full TS-IDM. Results on 10k datasets are shown in Table 3. We observe that the naïve autoencoder-based inverse dynamics module fails to provide reasonable representations.

Incorporating dynamics-related information via latent dynamics is helpful, but the performance gain remains mild. Enforcing ODE properties on decoders greatly enhances the quality of learned representations. Lastly, enforcing T-symmetry consistency proves to be the strongest performance improvement factor, which greatly enhances the quality of the learned representations for downstream policy learning.

Table 3: Ablation on the components of TS-IDM. The standard deviations are noted by $\pm$.

|  | Hopper-me | Halfcheetah-me | Walker2d-me |
|---|---|---|---|
| $\phi/\psi + h_{inv}$ | $17.2 \pm 7.0$ | $29.7 \pm 3.6$ | $24.5 \pm 10.1$ |
| $\uparrow + h_{fwd}, h_{rvs}$ | $35.5 \pm 7.3$ | $31.3 \pm 1.1$ | $33.6 \pm 9.2$ |
| $\uparrow + \ell_{\text{ode}}$ | $61.4 \pm 23.7$ | $31.2 \pm 1.2$ | $58.5 \pm 18.1$ |
| $\uparrow + \ell_{\text{T-sym}}$ | $\mathbf{100.9 \pm 6.8}$ | $\mathbf{40.7 \pm 1.2}$ | $\mathbf{87.4 \pm 13.3}$ |

**Effectiveness of the learned representations.**
As demonstrated in Figure 4(left), we further verify the effectiveness of the learned latent representation in TS-IDM. Specifically, we use TS-IDM's state encoder $\phi_s(s)$ as the representation learning module on top of two offline RL methods: IQL and TD3+BC. The results reveal significant performance improvements and variance reduction when IQL and TD3+BC are trained within the latent state space induced by $\phi_s(s)$, suggesting that TS-IDM learns compact and generalizable representations that benefit policy learning. To further evaluate the quality of TS-IDM's representations, in Figure 4(right), we replace TS-IDM in TELS with other representation learning methods, including autoencoder ("AE-rep"), variational autoencoder ("VAE-rep") (Kingma & Welling, 2014), and contrastive learning method SimCLR ("Contras-rep") (Chen et al., 2020). The results show that the TS-IDM representation achieves substantially better performance as compared to AE, VAE, and contrastive representations.

**Ablations on regularizer terms in policy optimization.** We also conduct ablation experiments in Figure 5 to validate the effectiveness of the T-symmetry consistency regularizer term $\ell_{\text{T-sym}}$ during the guide-policy optimization process of TELS. The results demonstrate that incorporating this term can effectively enhance performance while reducing variance, highlighting the importance of utilizing T-symmetry consistency regularization to promote OOD generalization and learning stability. We present more ablation experiment results in Section B.5.

## 5 RELATED WORK

Offline RL faces unique challenges in mitigating the risk of OOD exploitation. Evaluating value functions in OOD regions often results in inaccurate estimates, which can lead to severe value overestimation and misguiding policy learning. To mitigate this, most offline RL methods leverage data-related regularizations to stabilize the learning process. These include explicit behavior constraint techniques that penalize action divergence (Wu et al., 2019; Kumar et al., 2019; Fujimoto & Gu, 2021; Li et al., 2022; Liu et al., 2024), value regularization schemes to discourage policies from selecting OOD actions via modifying Bellman update (Kumar et al., 2020; Xu et al., 2022b; Bai et al., 2021; Lyu et al., 2022; Niu et al., 2022; 2025) or introducing uncertainty penalities (Wu et al., 2021; An et al., 2021; Bai et al., 2021), and in-sample learning methods (Brandfonbrener et al., 2021; Kostrikov et al., 2022; Xu et al., 2023; Mao et al., 2024b; Li et al., 2023; Wang et al., 2024; Zheng et al., 2024), which stabilize training by only using in-sample data for value and policy learning. While these methods perform reasonably well on datasets with sufficient state-action coverage, they often struggle in small-sample settings where exploiting OOD generalization is vital for achieving good performance. Recently, leveraging expressive model architectures such as Transformers and diffusion models (Wang et al., 2022; Ajay et al., 2022; Janner et al., 2022; Hansen-Estruch et al., 2023; Li et al., 2024; Mao et al., 2024a; Zheng et al., 2024; Liu et al., 2025) have gained popularity in offline RL, due to their strong capability to fit complex data distributions. However, these models are overly heavy and require extensive amounts of data to learn (Wang et al., 2025), making them hardly usable for the small-sample setting.

## 6 CONCLUSION

We propose a highly sample-efficient offline RL algorithm that learns an optimized policy within the latent space regulated by the fundamental T-symmetry property. Specifically, we develop a T-symmetry enforced inverse dynamics model, TS-IDM, to construct a well-behaved and generalizable latent space, effectively mitigating the challenges of OOD generalization. By learning a T-symmetry regularized guide-policy within this latent space, we can obtain the reward-maximizing next state to serve as the goal state input in the learned TS-IDM for optimal action extraction. Through extensive experiments, we show that TELS achieves strong OOD generalization capability and SOTA small-sample performance. Moreover, we empirically show that TS-IDM can also function as a representation learning model to provide informative representations and enhance the performance of existing methods under the small-sample setting. One potential limitation of TELS is that strong ODE and T-symmetry property regularizations, although helpful for capturing fundamental patterns in data, sometimes could limit the model's expressive power (see Appendix B.5). Furthermore, the current TELS framework is primarily optimized for deterministic dynamics. Future studies could explore improved designs to optimally balance fundamental pattern extraction with model expressivity and investigate the adaptation of TS-IDM to capture T-symmetry properties within stochastic environments.

## ACKNOWLEDGMENTS

This work is supported by GDS Holdings Limited and funding from Wuxi Research Institute of Applied Technologies, Tsinghua University under Grant 20242001120, and Xiongan AI Institute.

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

APPENDIX

# A    ADDITIONAL DISCUSSION ON RELATED WORKS

In this section, we present a detailed discussion of the connections and differences between our proposed method TELS with TSRL (Cheng et al., 2023), POR (Xu et al., 2022a), and conventional model-based offline RL approaches.

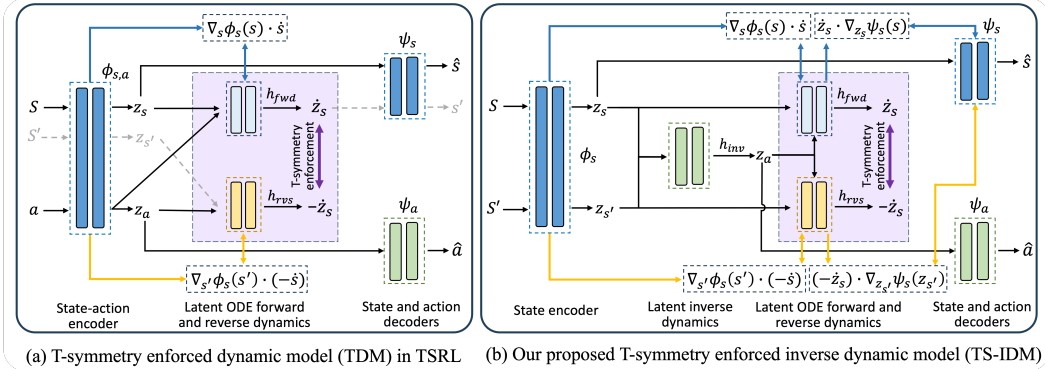

(a) T-symmetry enforced dynamic model (TDM) in TSRL      (b) Our proposed T-symmetry enforced inverse dynamic model (TS-IDM)

Figure 6: Comparison of the architecture between TDM in TSRL and our proposed TS-IDM in TELS.

**Connection and differences with TSRL.**    As illustrated in Figure 6, both TSRL and TELS leverage the T-symmetry consistency enforcement to construct the latent space. Specifically, in Figure 6(a), TSRL employs a T-symmetry-enforced dynamics model (TDM), which models system dynamics by incorporating paired latent ODE forward and reverse dynamics to enforce T-symmetry. In contrast, Figure 6(b) illustrates our proposed T-symmetry-enforced inverse dynamics model (TS-IDM), which integrates T-symmetry constraints into both forward and reverse dynamics while incorporating an inverse dynamics model. We emphasize the main differences between TELS and TSRL as follows:

- **Architecture:** As presented in Figure 6(a), TDM jointly encodes state-action pairs to form the latent space, which may capture behavioral biases from the dataset (e.g., expert-specific action patterns) and impede learning fundamental, distribution-agnostic dynamics patterns in data. In contrast, Figure 6(b) illustrates that TS-IDM overcomes these limitations by adopting a state-only modeling approach, focusing on the underlying latent state variations. Additionally, the only useful component of the learned TDM for downstream policy learning is its encoder $\phi(s, a)$, wasting the dynamics-related information captured by the model. In contrast, TS-IDM trains an inverse dynamics model within the T-symmetry-enforced latent space, which can be reused as an execute-policy to extract optimal actions.

- **Detailed model design:** As shown in Figure 6(a), TDM only enforces the ODE property for its encoder but not the decoder, which could lead to inconsistency between the learned dynamics and the underlying ODE structure, resulting in inaccurate or misaligned ODE representations. To address this problem, we introduce the loss term $\ell_{\mathrm{ode}}$ in Eq. (5) specifically to achieve this goal. This design is very important as it can greatly enhance the coupling among the different elements in the model and result in a more stable learning process.

- **Training procedure:** In TSRL, the TDM encoder and decoders must be pre-trained before joint training on other components to avoid stability issues. In contrast, our proposed TS-IDM does not require pre-training; all components can be learned jointly in a single stage. Additionally, TDM requires adding L1-norm regularization to the parameters of the latent forward and reverse dynamics models to stabilize the learning process. This is unnecessary in TS-IDM (see Eq. (7)), as the design of our proposed TS-IDM enables strongly coupled and consistent relationships among all its internal components. The learning curves of TS-IDM can be found in Appendix F.

- **Policy optimization:** Since TDM requires both state and action as inputs to derive the latent representations, it is constrained to Q-function maximization for policy optimization. Consequently, TSRL adopts the TD3+BC framework as its backbone for policy optimization, which inherently suffers from over-conservative action-level constraints, particularly in small dataset settings. In

contrast, TELS performs policy optimization entirely within the compact and generalizable latent state space derived from TS-IDM, enabling state-level optimization that avoids the limitations of action-space constraints.

**Connection and differences with POR.** As discussed in Section 2, while both POR and TELS share similarities in utilizing a state-stitching approach in state space for policy optimization, they exhibit the following fundamental differences:

- **Original state-space vs. latent state-space optimization:** POR relies on policy optimization in the original state space, which inherently requires sufficient state-action coverage for valid state-stitching. In contrast, TELS mitigates this limitation by constructing a compact and generalizable latent space via TS-IDM.

- **Unregularized T-symmetry vs. T-symmetry regularized policy optimization:** POR optimizes the guide-policy solely through an AWR formulation (Neumann & Peters, 2008; Peng et al., 2019), constraining $\pi_g$ to stay close to the dataset via state-stitching as in Eq. (2), but lacks additional regularization to ensure generalizable state transitions. In contrast, TELS enforces an additional T-symmetry consistency regularization $\ell_{\text{T-sym}}$, which plays a critical role in preventing $\pi_g$ from outputting problematic and non-generalizable latent next states, thereby enhancing its OOD generalizability.

**Naïvely combining TSRL and POR does not work.** Simply combining TSRL and POR actually performs notably worse than each method alone, as shown in Table 4. This performance degradation stems from a fundamental incompatibility between the TDM in TSRL and POR's state-stitching mechanism. In contrast, our proposed TELS successfully exploits both T-symmetry and state-stitching, leading to substantial improvements over all baselines.

**Differences from model-based approaches.** We emphasize that our proposed TELS framework fundamentally differs from MBRL methods (Janner et al., 2019; Kidambi et al., 2020; Yu et al., 2021b; Wang et al., 2021; Yu et al., 2021a; Zhan et al., 2022a;b; Rigter et al., 2022). Conventional MBRL methods prioritize learning forward dynamics models to predict future states and generate rollouts for policy learning. In contrast, our proposed TS-IDM is primarily designed for state representation learning and action extraction via inverse dynamics, rather than for data generation. Furthermore, as evidenced by Table 1, in the small-sample setting, limited data samples are insufficient for the model-based approach to learn an accurate dynamics model, causing high approximation errors during model rollouts, which significantly deteriorate policy learning performance.

Table 4: Performance comparison between TELS, TSRL, POR, and TSRL+POR on reduced-size D4RL datasets. The highest score in the row is bolded.

| Task | TELS | TSRL | POR | TSRL+POR |
|------|------|------|-----|----------|
| Hopper-m | **77.3 $\pm$ 10.7** | 62.0 $\pm$ 3.7 | 46.4$\pm$ 1.7 | 38.5 $\pm$ 2.4 |
| Hopper-mr | **43.2 $\pm$ 3.5** | 21.8 $\pm$ 8.2 | 17.4$\pm$ 6.2 | 25.9 $\pm$ 5.9 |
| Hopper-me | **100.9 $\pm$ 6.8** | 50.9 $\pm$ 8.6 | 37.9$\pm$ 6.1 | 30.3$\pm$9.7 |
| Halfcheetah-m | **40.8 $\pm$ 0.6** | 38.4 $\pm$ 3.1 | 33.3$\pm$ 3.2 | 35.2$\pm$ 7.5 |
| Halfcheetah-mr | **33.2 $\pm$ 1.0** | 28.1 $\pm$ 3.5 | 27.5$\pm$ 3.6 | 28.3$\pm$ 4.2 |
| Halfcheetah-me | **40.7 $\pm$1.2** | 39.9 $\pm$ 21.1 | 34.7$\pm$ 2.6 | 38.9 $\pm$ 1.6 |
| Walker2d-m | **62.4 $\pm$ 5.3** | 49.7 $\pm$ 10.6 | 22.2$\pm$ 3.6 | 25.7$\pm$ 16.9 |
| Walker2d-mr | **54.8 $\pm$ 6.0** | 26.0 $\pm$ 11.3 | 14.8$\pm$4.2 | 12.9$\pm$ 3.2 |
| Walker2d-me | **87.4 $\pm$ 13.3** | 46.4 $\pm$ 17.4 | 20.1$\pm$ 8.6 | 23.8$\pm$ 9.8 |
| Antmaze-u | **88.7 $\pm$ 7.7** | 76.1 $\pm$ 15.6 | 42.1$\pm$ | 40.4$\pm$18.1 |
| Antmaze-u-d | **60.9 $\pm$ 16.9** | 52.2 $\pm$ 22.1 | 6.1$\pm$ | 6.7$\pm$3.1 |
| Antmaze-m-d | **47.2 $\pm$ 17.3** | 0.0 | 0.0 | 0.0 |
| Antmaze-m-p | **62.9 $\pm$ 17.8** | 0.0 | 0.0 | 0.0 |
| Antmaze-l-d | **39.8 $\pm$ 14.1** | 0.0 | 0.0 | 0.0 |
| Antmaze-l-p | **47.3 $\pm$ 13.1** | 0.0 | 0.0 | 0.0 |

# B  ADDITIONAL RESULTS

## B.1  EVALUATION ON THE FULL DATASETS

We also evaluate the performance of TELS on the original full datasets of D4RL tasks, and the results are presented in Table 5. Our proposed method achieves comparable or better performance than existing offline RL methods. Note that although TSRL also adopts a similar T-symmetry regularized representation learning scheme as ours, it performs poorly in Antmaze medium and large datasets. Primarily due to its use of the conservative TD3+BC backbone for policy optimization.

Table 5: Normalized scores on full-size D4RL datasets (averaged over the final 10 evaluations with 5 seeds). The highest score in the row is bolded.

| Task | BC | TD3+BC | MOPO | COMBO | CQL | IQL | DOGE | IDQL | POR | TSRL | TELS (ours) |
|---|---|---|---|---|---|---|---|---|---|---|---|
| Hopper-m | 52.9 | 59.3 | 28.0 | 97.2 | 58.5 | 66.3 | **98.6 ± 2.1** | 63.1 | 78.6 ± 7.2 | 86.7±8.7 | 94.3 ± 2.8 |
| Hopper-mr | 18.1 | 60.9 | 67.5 | 89.5 | 95.0 | 94.7 | 76.2±17.7 | 82.4 | 98.9 ± 2.1 | 78.7±28.1 | **99.5 ± 2.3** |
| Hopper-me | 52.5 | 98.0 | 23.7 | **111.1** | 105.4 | 91.5 | 102.7± 5.2 | 105.3 | 90.0 ± 12.1 | 95.9±18.4 | 105.4 ± 8.5 |
| Halfcheetah-m | 42.6 | 48.3 | 42.3 | **54.2** | 44.0 | 47.4 | 45.3± 0.6 | 49.7 | 48.8 ± 0.5 | 48.2 ±0.7 | 44.3 ± 0.4 |
| Halfcheetah-mr | 36.6 | 44.6 | 53.1 | 55.1 | 45.5 | 44.2 | 42.8 ±0.6 | **45.1** | 43.5±0.9 | 42.2 ± 3.5 | 41.1 ± 0.1 |
| Halfcheetah-me | 55.2 | 90.7 | 63.3 | 90.0 | 91.6 | 86.7 | 78.7±8.4 | 94.4 | **94.7±2.2** | 92.0±1.6 | 93.1 ± 1.5 |
| Walker2d-m | 75.3 | 83.7 | 17.8 | 81.9 | 72.5 | 78.3 | **86.8 ± 0.8** | 80.2 | 81.1 ± 2.3 | 77.5 ±4.5 | 81.3± 5.1 |
| Walker2d-mr | 26.0 | 81.8 | 39.0 | 56.0 | 77.2 | 73.9 | **87.3 ± 2.3** | 79.8 | 76.6 ± 6.9 | 66.1±12.0 | 86.0 ± 3.3 |
| Walker2d-me | 107.5 | 110.1 | 44.6 | 103.3 | 108.8 | 109.6 | 110.4±1.5 | **111.6** | 109.1 ± 0.7 | 109.8±3.1 | 110.7 ± 1.4 |
| Antmaze-u | 65.0 | 78.6 | 0.0 | 80.3 | 84.8 | 85.5 | **97.0 ± 1.8** | 93.8 | 90.6 ± 7.1 | 81.4 ± 19.2 | 94.5 ± 10.3 |
| Antmaze-u-d | 45.6 | 71.4 | 0.0 | 57.3 | 43.4 | 66.7 | 63.5 ± 9.3 | 62.0 | 71.3 ± 12.1 | 76.5 ± 29.7 | **79.7 ± 15.3** |
| Antmaze-m-d | 0.0 | 0.0 | 0.0 | 0.0 | 54.0±11.7 | 74.6±3.2 | 77.6±6.1 | **86.6** | 79.2±3.1 | 0.0 | 82.4 ± 4.5 |
| Antmaze-m-p | 0.0 | 0.0 | 0.0 | 0.0 | 65.2±4.8 | 70.4±5.3 | 80.6±6.5 | 83.5 | 84.6 ±5.6 | 0.0 | **86.7 ± 5.7** |
| Antmaze-l-d | 0.0 | 0.0 | 0.0 | 0.0 | 31.6±9.5 | 45.6±7.6 | 36.4 ±9.1 | 56.4 | 73.4 ±8.5 | 0.0 | **75.7 ± 11.2** |
| Antmaze-l-p | 0.0 | 0.0 | 0.0 | 0.0 | 18.8±15.3 | 43.5±4.5 | 48.2±8.1 | 57.0 | 58.0 ± 12.4 | 0.0 | **60.7 ± 13.3** |

## B.2  ADDITIONAL RESULTS ON ADROIT TASKS

We conduct additional experiments on Adroit-cloned/expert tasks. Since these tasks have much larger datasets (500k) as compared to Adroit-human tasks (5k samples), substantially reducing the learning difficulty, we therefore test our methods against baselines on a more challenging reduced-size setting with 10k samples. The results in Table 6 demonstrate TELS still achieves strong performance.

Table 6: Performance comparison of TELS against baseline algorithms on Adroit tasks with limited data (10k). Numbers at or above 95% of the best in the row are highlighted in bold.

| Task | Size (ratio) | BC | TD3+BC | MOPO | COMBO | CQL | IQL | DOGE | IDQL | POR | TSRL | TELS |
|---|---|---|---|---|---|---|---|---|---|---|---|---|
| Pen-cloned | 10k (2%) | 37.4 ± 37.6 | 0.1 ± 3.0 | 0.1 ± 0.1 | 0.7 ± 0.2 | 1.5 ± 4.8 | 35.6 ± 30.5 | 30.1 ± 19.7 | 64.4 ± 15.1 | 43.6 ± 5.8 | 41.6 ± 27.5 | **69.7 ± 12.6** |
| Pen-expert | 10k (2%) | 27.6 ± 21.3 | 5.2 ± 2.7 | 1.2 ± 0.3 | 2.5 ± 0.4 | 3.6 ± 4.5 | 68.9 ± 24.3 | 31.1 ± 19.3 | **104.6 ± 3.8** | 61.2 ± 21.0 | 65.6 ± 22.8 | **105.7 ± 12.1** |
| Hammer-cloned | 10k (2%) | 0.3 ± 0.4 | 0.2 ± 0.1 | 0.1 ± 0.1 | 0.2 ± 0.1 | 0.2 ± 0.1 | 0.4 ± 0.2 | 0.3 ± 0.1 | **0.8 ± 0.3** | 0.1 ± 0.1 | 0.6 ± 0.3 | 0.6 ± 0.2 |
| Hammer-expert | 10k (2%) | 0.2 ± 0.1 | 0.5 ± 0.2 | 0.1 ± 0.1 | 0.2 ± 0.1 | 1.2 ± 1.1 | 70.3 ± 30.3 | 0.6 ± 0.3 | **91.7 ± 12.9** | 2.7 ± 2.6 | 77.6 ± 31.2 | **91.5 ± 25.9** |
| Door-cloned | 10k (2%) | 0.1 ± 0.1 | 0.3 ± 0.1 | 0.2 ± 0.1 | 0.1 ± 0.3 | 0.2 ± 0.1 | 1.5 ± 0.8 | 0.5 ± 0.5 | 0.1 ± 0.1 | 0.1 ± 0.1 | 0.1 ± 0.3 | **7.6 ± 2.3** |
| Door-expert | 10k (2%) | 1.2 ± 1.1 | 5.2 ± 3.1 | 1.5 ± 1.2 | 3.5 ± 1.1 | 20.3 ± 15.7 | 79.2 ± 8.8 | 0.5 ± 0.1 | **98.3 ± 5.5** | 0.7 ± 0.3 | 46.3 ± 12.5 | **101.8 ± 8.5** |
| Relocate-cloned | 10k (2%) | 0.2 ± 0.1 | 0.3 ± 0.1 | 0.3 ± 0.2 | 0.1 ± 0.1 | 0.3 ± 0.1 | 0.1 ± 0.5 | 0.1 ± 0.1 | **0.2 ± 0.2** | 0.1 ± 0.1 | 0.2 ± 0.1 | 0.2 ± 0.1 |
| Relocate-expert | 10k (2%) | 0.6 ± 0.1 | 0.1 ± 0.1 | 0.1 ± 0.2 | 1.5 ± 1.2 | 0.2 ± 0.1 | 31.1 ± 8.4 | 0.3 ± 0.5 | **87.5 ± 12.7** | 0.2 ± 0.1 | 45.2 ± 15.3 | **85.6 ± 12.1** |

## B.3  ADDITIONAL OOD GENERALIZABILITY VALIDATION EXPERIMENTS

We further investigate the generalization capabilities of DOGE (Li et al., 2022), IDQL (Hansen-Estruch et al., 2023), and TSRL (Cheng et al., 2023) under the variation deletion degrees in the Antmaze environment. Specifically, we train each algorithm on the modified dataset after the deletion operation. We then evaluate their behaviors by visualizing rollouts over 20 evaluation episodes.

As illustrated in Figure 7, only IDQL occasionally succeeds in reaching the goal under the 0% deletion setting, while both DOGE and TSRL fail consistently. As the deletion ratio increases to 70% and 100%, none of the three methods achieves meaningful policy learning. These results highlight the inherent challenges of this setting, which requires both a compact yet expressive latent representation

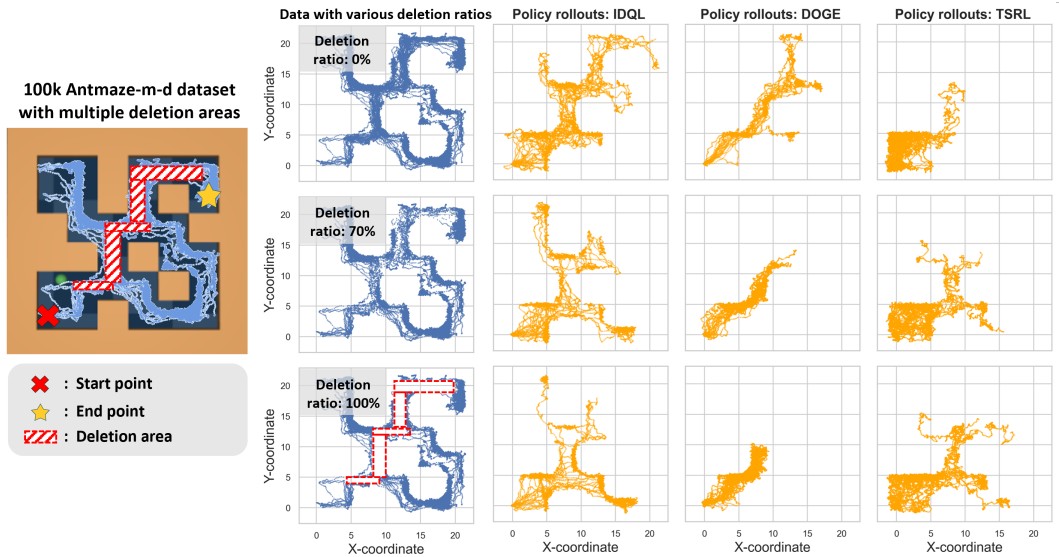

Figure 7: **Left:** Illustration of the 100k Antmaze-m-d task with multiple deletion areas, where the red cross denotes the start point, the yellow star denotes the goal locations, and the red shaded areas denote the data deletion regions. **Right:** Visualization of the training dataset and policy rollout trajectories generated by trained policies from various algorithms under varying deletion ratios.

space and a highly generalizable policy capable of operating with extremely sparse and limited data. While TSRL integrates TDM to distill underlying patterns from the dataset, the scarcity of available data undermines its action-level constraints approach, preventing it from deriving a viable policy.

## B.4 ADDITIONAL RLIABLE PLOTS

To further statistically justify the performance of TELS, we use Rliable (Agarwal et al., 2021b) to plot the aggregate results across all locomotion tasks with 10k avaliable dataset. As shown in Figure 8, the results demonstrate that TELS consistently yields the highest score with the minimal optimality gap compared to all baselines.

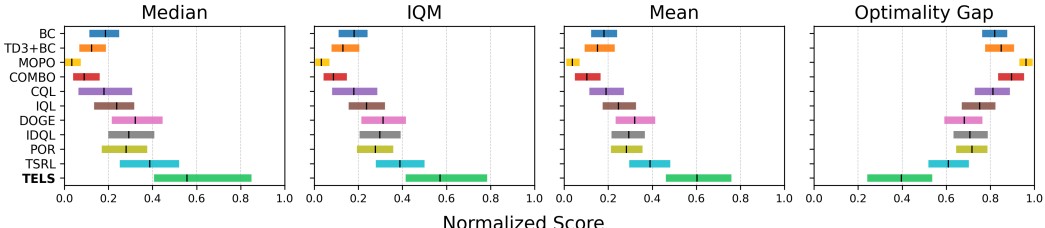

Figure 8: Rliable plots for locomotion tasks with 10k dataset over 5 random seeds.

## B.5 ADDITIONAL ABLATION EXPERIMENTS

**Ablations of $\beta$ in TS-IDM.** We find that a good weighting parameter $\beta$ value typically corresponds to low TS-IDM training loss Eq. (7). As shown in Figure 9, $\beta = 0.1$ yields the lowest loss for HalfCheetah, while $\beta = 1$ is better for Hopper and Walker2d. Table 7 further confirms that the same $\beta$ setting with $\beta = 1$ gives the highest score on Hopper and Walker2d, whereas $\beta = 0.1$ performs best on HalfCheetah. This shows that the better the TS-IDM is trained, the higher quality latent representation can be learned to facilitate downstream policy optimization. This is important, as it indicates that we do not require any environmental interaction or policy evaluation for $\beta$ selection.

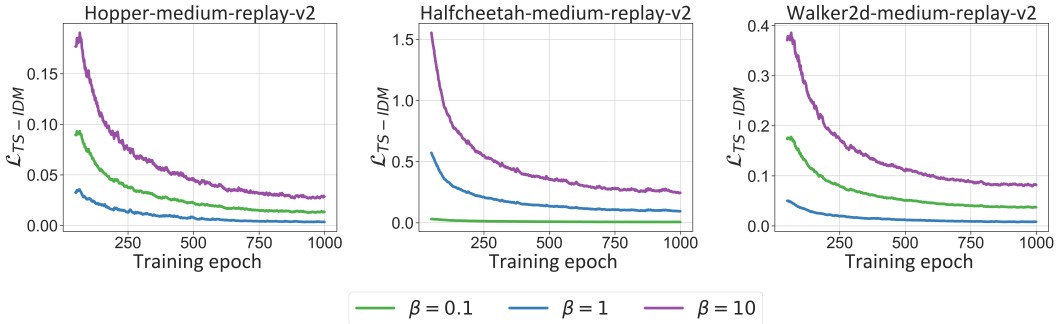

Figure 9: The learning curves for training TS-IDM on 10k dataset with different $\beta$ hyperparameter. The standard deviations are noted by $\pm$. Numbers at or above 95% of the best in the row are highlighted in bold.

Table 7: TELS performance with different $\beta$ under small-sample setting. Numbers at or above 95% of the best in the column are highlighted in bold.

|  | $\beta = 10$ | $\beta = 1$ (Used) | $\beta = 0.1$ |
|---|---|---|---|
| Hopper-m | $77.3 \pm 5.4$ | $\mathbf{77.3 \pm 10.7}$ | $61.4 \pm 5.6$ |
| Hopper-mr | $15.3 \pm 6.6$ | $\mathbf{43.2 \pm 3.5}$ | $19.7 \pm 3.4$ |
| Hopper-me | $37.6 \pm 17.9$ | $\mathbf{100.9 \pm 6.8}$ | $64.7 \pm 3.3$ |
| Halfcheetah-m | $32.9 \pm 2.3$ | $40.8 \pm 0.6$ | $\mathbf{41.2 \pm 1.1}$ |
| Halfcheetah-mr | $8.6 \pm 1.8$ | $33.2 \pm 1.0$ | $\mathbf{34.0 \pm 2.2}$ |
| Halfcheetah-me | $7.5 \pm 2.2$ | $40.7 \pm 1.2$ | $\mathbf{41.5 \pm 2.1}$ |
| Walker2d-m | $37.2 \pm 7.9$ | $\mathbf{62.4 \pm 5.3}$ | $54.6 \pm 8.2$ |
| Walker2d-mr | $17.1 \pm 2.9$ | $\mathbf{54.8 \pm 6.0}$ | $39.2 \pm 8.6$ |
| Walker2d-me | $20.4 \pm 10.4$ | $\mathbf{87.4 \pm 13.3}$ | $44.7 \pm 9.8$ |

Table 8: TELS performance under various weighting terms on 10k Hopper-me dataset.

| Weight of $\ell_{\mathbf{dyn}}$ | Weight of $\ell_{\mathbf{ode}}$ | Weight of $\ell_{\mathbf{T\text{-}sym}}$ | Evaluation scores |
|---|---|---|---|
| **1** | **1** | **1** | $\mathbf{100.9 \pm 6.8}$ (Used) |
| 1 | 1 | 0.1 | $56.4 \pm 0.6$ |
| 1 | 0.1 | 1 | $51.1 \pm 2.2$ |
| 0.1 | 1 | 1 | $56.2 \pm 3.1$ |
| 1 | 0.1 | 0.1 | $56.3 \pm 3.2$ |
| 0.1 | 1 | 0.1 | $54.8 \pm 11.4$ |
| 0.1 | 0.1 | 1 | $56.1 \pm 5.7$ |

Simply looking at the supervised training loss of TS-IDM on offline datasets will already provide a good sense of the proper scale.

Empirically, we observe that smaller datasets benefit from a relatively larger $\beta$, whereas in large datasets, a smaller $\beta$ is typically required to reduce training loss. This is as expected, as $\beta$ controls the strength of T-symmetry and ODE regularization. Large datasets contain sufficient information from data samples, thus requiring less regularization, while small datasets benefit from stronger regularization to enable the extraction of additional information from limited samples. To keep it simple, in our main results, we use $\beta = 1$ for all MuJoCo locomotion tasks in the small-sample setting without hyperparameter tuning. For tasks with large datasets and other domains, we select $\beta$ from the set $\{0.01, 0.1, 1\}$ as the one with the lowest TS-IDM training loss.

Furthermore, as we have discussed in the main paper, we need to use a single shared $\beta$ for loss terms $\ell_{\text{dyn}}$, $\ell_{\text{ode}}$, and $\ell_{\text{T-sym}}$. To provide some evidence, we also conducted an extra ablation experiment by re-weighting each term in Eq. (7) differently as in Table 8. The results confirm that inconsistent $\beta$ weighting schemes lead to significant degradation in policy performance and unstable learning, ultimately resulting in poor outcomes. By contrast, simply using the same $\beta$ value achieves doubled evaluation scores. The reason behind this is what we have explained in the main paper, the internal components of TS-IDM are strongly coupled and have to be regulated at the same strength (i.e., using the same $\beta$). Specifically, both the latent ODE forward and reverse dynamics modules ($h_{fwd}, h_{rvs}$) use the same latent actions $z_a$ from $h_{inv}$ as input. The T-symmetry consistency $\ell_{\text{T-sym}}$ is also enforced on both $h_{fwd}, h_{rvs}$. These make the latent forward, reverse, and inverse dynamics modules strongly coupled. Moreover, the state encoder $\phi_s$ and decoder $\psi_s$ also need to satisfy the ODE property as required in the $h_{fwd}$ and $h_{rvs}$, as enforced through the loss terms $\ell_{\text{dyn}}$ and $\ell_{\text{ode}}$ respectively. Hence, if different levels of regularizations are applied to these loss terms, internal inconsistency will emerge and impair the learning of TS-IDM.

**Impact of regularizer terms $\eta$ in policy optimization.** The hyperparameter $\eta$ governs the strength of regularization in TELS, balancing exploration and adherence to dataset states during policy updates. To evaluate the robustness of TELS, we test multiple $\eta$ values ($\eta = \{1, 5, 10\}$) to examine its sensitivity to the state-level behavioral constraint in Eq. (9). Higher $\eta$ values impose stronger constraints on the guide-policy, requiring generated states $s'$ to align closely with dataset states. As shown in Figure 10, TELS demonstrates consistent robustness across $\eta$ settings, achieving reliable performance under varying constraint strengths.

**Impact of each component in TS-IDM for policy optimization.** To further validate the impact of the T-symmetry regularizer $\ell_{\text{T-sym}}$ in Eq. (10), we conduct additional ablation studies on 100k-sample Antmaze tasks. From the evaluation results presented in Table 9, the naïve auto-encoder based inverse dynamics module "$\phi/\psi + h_{inv}$" fails to form a reasonable latent space, yielding 0 average normalized scores across all Antmaze tasks. The introduction of latent dynamics models "$h_{fwd}$" and "$h_{rvs}$" provides marginal improvements by capturing partial system dynamics, yet remains insufficient for effective policy learning. Notably, enforcing ODE properties on decoders and applying T-symmetry consistency emerge as the most significant factors driving performance improvements, substantially enhancing the reliability of learned representations for downstream guide-policy optimization.

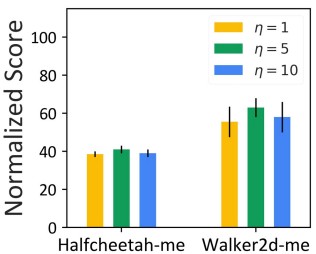

Figure 10: TELS with various $\eta$.

Table 9: Ablations on the components of TS-IDM with 100k Antmaze datasets. Numbers at or above 95% of the best in the row are highlighted in bold.

|  | Antmaze-m-d | Antmaze-m-p | Antmaze-l-d | Antmaze-l-p |
|---|---|---|---|---|
| $\phi/\psi + h_{inv}$ | 0 | 0 | 0 | 0 |
| $\uparrow + h_{fwd}, h_{rvs}$ | $23.6 \pm 18.4$ | $30.4 \pm 9.3$ | $14.4 \pm 5.6$ | $7.8 \pm 3.4$ |
| $\uparrow + \ell_{\text{ode}}$ | $34.1 \pm 15.7$ | $48.7 \pm 13.3$ | $20.1 \pm 8.9$ | $22.6 \pm 16.7$ |
| $\uparrow + \ell_{\text{T-sym}}$ | $\mathbf{47.2 \pm 17.3}$ | $\mathbf{62.9 \pm 17.8}$ | $\mathbf{39.8 \pm 14.1}$ | $\mathbf{47.3 \pm 13.1}$ |

**Impact of T-symmetry regularizer term in guide-policy optimization with stochastic policy instantiation.** We further conduct ablation experiments in Figure 11(left) to validate the effectiveness of the T-symmetry consistency regularization term $\ell_{\text{T-sym}}$ during the stochastic guide-policy optimization process of TELS. The results demonstrate that in stochastic policy optimization schemes, integrating this term significantly improves performance while reducing variance, underscoring the critical role of T-symmetry consistency regularization in enhancing OOD generalization and training stability.

**Effectiveness of learned representations for guide-policy optimization with stochastic policy instantiation.** As illustrated in Figure 11(right), we evaluate TELS across diverse representation learning approaches in Antmaze tasks. The results demonstrate that baseline models struggle to construct meaningful latent spaces as task complexity increases and data scarcity intensifies (with only 100k usable samples). In contrast, TS-IDM uniquely learns a compact, well-structured latent space that remains informative and generalizable, providing a more reliable latent space for policy learning.

## B.6 VISUAL ANALYSIS ON THE LEARNED LATENT SPACE

**Effectiveness of learned latent state space.** To illustrate the compactness and effectiveness of the learned latent state space through TS-IDM, in Figure 12, we plot the t-SNE visualization of the original data trajectories of the Hopper-m 10k task, as well as the rollout trajectories of learned TELS and IQL policies on both the original state space and the latent state space (encoded using our TS-IDM state encoder). We can clearly observe that the learned latent space is much more compact and well-behaved. The policy rollout trajectories form clear, continuous line patterns in our learned latent space, but can be quite noisy in the original state space. Such a more compact and structured latent space greatly facilitates robust policy learning via latent stitching and OOD generalization.

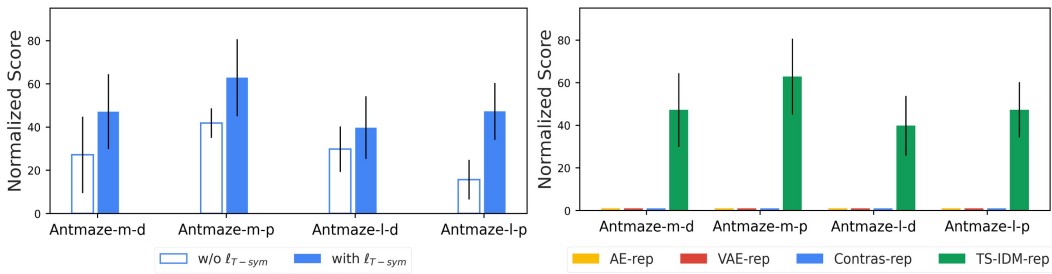

Figure 11: **Left:** Impact of $\ell_{\text{T-sym}}$ on policy optimization with 100k Antmaze datasets. **Right:** Performance of TELS with different representation models on Antmaze 100k datasets.

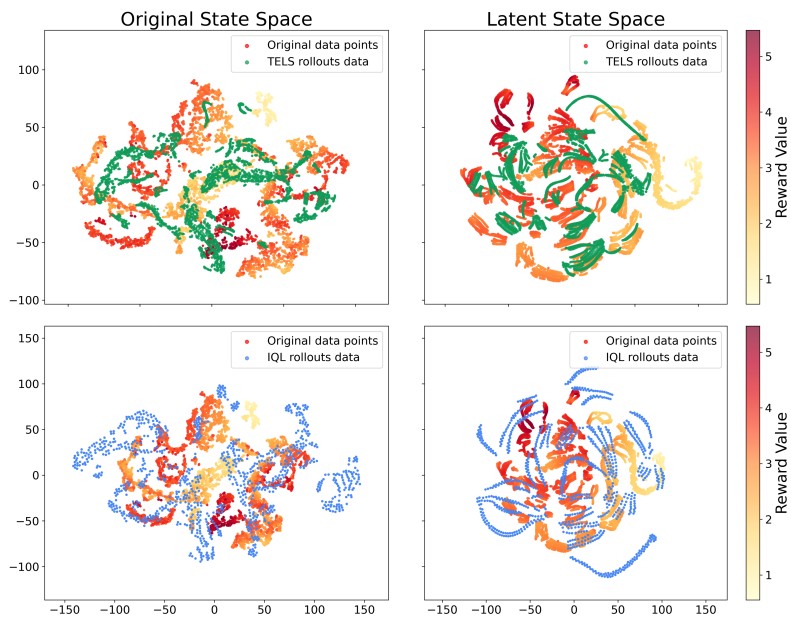

Figure 12: The t-SNE visualization of state representations on the Hopper-medium 10k task. The plots compare the original state space (left) with the latent space encoded by the pre-trained TS-IDM encoder $\phi_s$ (right). The data trajectory samples are colored by reward value. Overlaid points represent rollout trajectories (two episodes of 1,000 steps each) generated by the TELS policy (green) and the IQL policy (blue).

From the t-SNE visualization, we can observe that the IQL's rollout trajectories deviate substantially from the data distribution, generating numerous OOD states that violate the offline dataset distribution boundaries. Even when projected through the pre-trained TS-IDM encoder, these irrational states remain outside the meaningful latent manifold, elucidating the primary cause of IQL's performance degradation in data-scarce scenarios. In contrast, TELS demonstrates superior state-space utilization by maintaining good alignment with the dataset distribution while effectively navigating toward high-reward regions.

# C  IMPLEMENTATION DETAILS

## C.1  ALGORITHM PSEUDOCODE

The pseudocode of TELS is listed in Algorithm 1.

---

**Algorithm 1** Offline RL via T-symmetry Enforced Latent State-Stitching (TELS).

---

**Require:** Offline dataset $\mathcal{D}$.

1: *// TS-IDM learning*
2: Learning the state encoder $\phi_s$, state decoder $\psi_s$, action decoder $\psi_a$, latent inverse dynamics $h_{inv}$, latent forward and reverse dynamics $h_{fwd}$ and $h_{rvs}$ using the TS-IDM learning objective Eq. (7).
3: Initialize $V_\theta, V_{\theta'}, \pi_\sigma$
4: *// Policy training*
5: **for** $t = 1, \cdots, M$ training steps **do**
6:     Sample transitions $(s, r, s') \sim \mathcal{D}$ and compute their representations $(z_s, z_{s'})$ using the state encoder $\phi_s$.
7:     Use $(z_s, r, z_{s'})$ to update the latent state-value function $V$ using Eq.(8).
8:     Use $(z_s, z_{s'})$ to update the latent guide-policy $\pi_g$ using Eq. (9) or (10).
9: **end for**
10: *// Evaluation*
11: Get initial state $s$ from environment
12: **while** not done **do**
13:     Get optimized next state $z_{s'}^*$ using guide-policy $\pi_g$.
14:     Extract action $a$ using Eq. (11).
15: **end while**

---

## C.2   IMPLEMENTATION DETAILS OF TS-IDM

**Network structure.** For all MuJoCo locomotion and Antmaze tasks, we deployed 3-layer feed-forward neural networks for the state encoder $\phi_s$, latent inverse dynamics module $h_{inv}$, forward and reverse dynamics models $h_{fwd}$ and $h_{rvs}$, and decoder models $\psi_s$ and $\psi_a$ for the latent states and actions. The activation function is ReLU and uses the Adam optimizer to update the parameters. We present the hyperparameter details of training TS-IDM in Table 10, including the details of the structure we have implemented as well as the deployed hyperparameters.

**ODE property enforcement on $\phi_s$ and $\psi_s$.** We adopt a similar approach to TSRL (Cheng et al., 2023) to train the ODE enforced forward and reverse dynamic models. Specifically, we compute the time-derivative of the state encoder $\phi_s(s)$ by calculating its Jacobian matrix through `vmap()` function in Functorch [†]. This allows us to derive the supervision values $\frac{d\phi_s(s)}{ds} \cdot \dot{s}$ and $\frac{d\phi_s(s')}{ds'} \cdot (-\dot{s})$ for the forward dynamics module and reverse dynamics module respectively as in Eq. (4). This approach implicitly enforces the ODE property on the state encoder $\phi_s$ as the encoder is required to produce state representations that satisfy the ODE constraints. Unlike TSRL, which enforces ODE properties only on the encoders and not on the decoders, our method further regularizes the state decoder $\psi_s$. Specifically, $\psi_s$ is trained to decode the predicted latent state variables generated by $h_{fwd}(z_s, z_a) = \dot{z}_s$ and $h_{rvs}(z_{s'}, z_a) = -\dot{z}_s$ ensuring that it also satisfies the ODE constraints in Eq. (5). To achieve this, we apply the same approach to compute $\frac{d\psi_s(z_s)}{dt}$ and train the state decoder accordingly.

## C.3   IMPLEMENTATION DETAILS OF T-SYMMETRY REGULARIZED GUIDE-POLICY

**Network structure.** For all D4RL MuJoCo-v2 and Antmaze-v1 tasks, we deployed 2-layer feed-forward neural networks for the guide-policy $\pi_g$ and the value function $V$. The activation function is ReLU and uses the Adam optimizer to update the parameters. The parameter details are presented in Table 11.

**Hyperparameters for policy optimization.** Under both small-sample and full datasets settings, we employ a deterministic policy update strategy for MuJoCo locomotion tasks, as defined in Eq. (9), with learning rates of 1e-4 for both value and policy functions. The normalization term $\lambda$ is computed as $\lambda_\alpha = \alpha / [\sum_{s_i} |V(\phi_s(s_i))| / N]$, where $\alpha$ controls the trade-off between value maximization and policy regularization and $N$ denotes the number of samples in the training batch. For Antmaze tasks, we employ a stochastic policy optimization strategy, as outlined in Eq. (10), with learning rates of 1e-3 for both the value and policy functions.

---

[†]https://pytorch.org/functorch/stable/functorch.html

Table 10: Hyperparameters of TS-IDM.

| | Hyperparameters | Value |
|---|---|---|
| | State encoder hidden units | $512 \times 256$ |
| | State encoder activation function | ReLU |
| | Latent forward module hidden units | $256 \times 256$ |
| | Latent forward module activation function | ReLU |
| | Latent reverse module hidden units | $256 \times 256$ |
| | Latent reverse module activation function | ReLU |
| TS-IDM | latent inverse module hidden units | $1024 \times 1024$ |
| Architecture | Latent inverse module activation function | ReLU |
| | Latent inverse module dropout | True |
| | Latent inverse module dropout rate | 0.1 |
| | State decoder hidden units | $256 \times 512$ |
| | State decoder activation function | ReLU |
| | Action decoder hidden units | $512 \times 512$ |
| | Action decoder activation function | ReLU |
| | Optimizer type | Adam |
| | Weight of $\ell_{rec}$ | 1 |
| | Learning rate | 3e-4 |
| | Batch size | 256 |
| Training | Training epoch | 1000 |
| Parameters | State normalize | True |
| | Weight of $\beta$ | Selected from {0.01,0.1,1} as the one with the lowest TS-IDM training loss (see Figure 9) |
| | Weight decay | 0 (MuJoCo locomotion 10k setting)
1e-5 (Other tasks) |

Table 11: Structure and training parameters of guide-policy optimization.

| | Hyperparameters | Value |
|---|---|---|
| | Value network hidden units | $1024 \times 1024$ |
| Guide-policy | Value network activation function | ReLU |
| structure | Policy network hidden units | $1024 \times 1024$ |
| | Policy network hidden units | ReLU |
| | Optimizer type | Adam |
| | Target Value network moving average | 0.05 |
| | Batch size | 256 |
| | Training steps | 100,000 |
| Training | State normalize | True |
| Perparameters | Weight of $\tau$ | 0.9 (Antmaze tasks)
0.7 (Other tasks) |
| | Weight of $\alpha$ | 10 (Antmaze tasks)
0.01 (Other tasks) |

## C.4 MODEL COMPLEXITY AND TRAINING TIME.

As we presented the model structure details in Table 10, TS-IDM is actually a relatively small model, consisting of only 2-layer MLP sub-modules. Its parameter size (~2.8M parameters) is significantly smaller compared to many recent Transformer-based (~12M parameters) and diffusion-based (~16M parameters) offline RL methods. To further demonstrate the learning speed of TELS, we present a comparative analysis of training times with other baseline methods on the hopper-medium-v2 10k dataset, utilizing the official codebases. All the algorithms are trained on a workstation with an AMD Ryzen 9 7950X 16-Core Processor, NVIDIA GeForce RTX 4090 GPU, and 16GB of memory, running on Ubuntu 22.04.2 LTS 64-bit. As illustrated in Table 12, the Jax implementation of TELS completes training in merely 20 minutes, whereas the PyTorch version requires 120 minutes. This

result not only matches but often surpasses the efficiency of most baseline methods, underscoring the exceptional training efficiency of the method.

Table 12: Training time cost comparison on 10k Hopper-m datasets across various algorithms.

| Algorithm | Dynamics model training (min) | Policy optimization (min) | Total run time (min) | Evaluation scores |
|---|---|---|---|---|
| TELS (JAX) | 5 | 15 | 20 | **75.2 ± 6.3** |
| TELS (PyTorch) | 20 | 100 | 120 | **77.3 ± 10.7** |
| TSRL (github.com/pcheng2/TSRL) | 30 | 130 | 160 | 62.0 ± 3.7 |
| POR (github.com/ryanxhr/POR) | - | 450 | 450 | 46.4 ± 1.7 |
| IDQL (github.com/philippe-eecs/IDQL) | - | 470 | 470 | 44.2 ± 12.1 |
| DOGE (github.com/Facebear-ljx/DOGE) | - | 410 | 410 | 44.2 ± 10.2 |
| IQL (github.com/ikostrikov/implicit_q_learning) | - | 50 | 50 | 46.7 ± 6.5 |
| CQL (github.com/aviralkumar2907/CQL) | - | 780 | 780 | 43.1 ± 24.6 |
| COMBO (github.com/Shylock-H/COMBO_Offline_RL) | - | 1200 | 1200 | 30.2 ± 28.0 |
| MOPO (github.com/junming-yang/mopo) | - | 780 | 780 | 5.5 ± 2.3 |
| TD3+BC (github.com/sfujim/TD3BC) | - | 240 | 240 | 40.1 ± 18.6 |
| BC | - | 100 | 100 | 29.7 ± 11.7 |

# D  DETAILED EXPERIMENT SETUPS

## D.1  EXPERIMENT SETUP FOR SIMULATION BENCHMARK TASKS

**Reduced-size dataset generation.**  To create reasonably reduced-size D4RL datasets for a fair comparison, we use the identical small samples as in the TSRL paper (Cheng et al., 2023) for the locomotion tasks training. For Antmaze tasks, we adopt a similar approach by randomly sub-sampling trajectories from the original dataset to construct smaller training datasets. Specifically, for the "Antmaze-umaze" tasks, we randomly sample 10k data points for training, and for the "Antmaze-medium" and "Antmaze-large" tasks, we utilize 100k random samples as the training dataset of TELS.

The rationale behind this adjustment is the "medium" and "large" environments are significantly more expansive than the "umaze" environment. Sampling only 10k data points would likely result in trajectories that lack the fundamental information necessary to describe the task. Therefore, we relax the small-sample constraints for these environments to ensure that the reduced datasets at least contain enough successful trajectories for effective training.

**Experiment setups for various representation learning.**  To validate the effectiveness of the representations learned by TS-IDM, we integrate it as the representation module in two offline RL frameworks (IQL and TD3+BC), verifying the usability of the learned latent space as illustrated in Figure 4(left). Specifically, we process the original states $s$ and next states $s'$ from the dataset using the pre-trained state encoder $\phi_s$ of TS-IDM to derive the latent representations: $\phi_s(s) \to z_s$ and $\phi_s(s') \to z_{s'}$. Then, train IQL and TD3+BC within the latent space to evaluate their performance under the small-sample setting.

Furthermore, in Figure 4(right), we benchmark TELS against three established representation learning baselines ("AE-rep", "Contras-rep" and "VAE-rep") to rigorously assess TS-IDM's representation quality. Implementation details for all baseline models are provided below:

• **"AE-rep"**: We implement a naïve autoencoder-based inverse dynamics framework, consisting of a state encoder and decoders $\phi_s$ and $\psi_s$ to construct the latent state space. As in TELS, the inverse dynamics model $h_{inv}$ is built within this latent space, serving as the execute-policy. For a fair comparison, we use the same network parameters for the encoder, decoder, and inverse dynamics module as in TS-IDM. The "AE-rep" model is trained with a reconstruction loss to capture the essential features of the input, and the inverse dynamics model is simultaneously trained on the latent representations to predict actions.

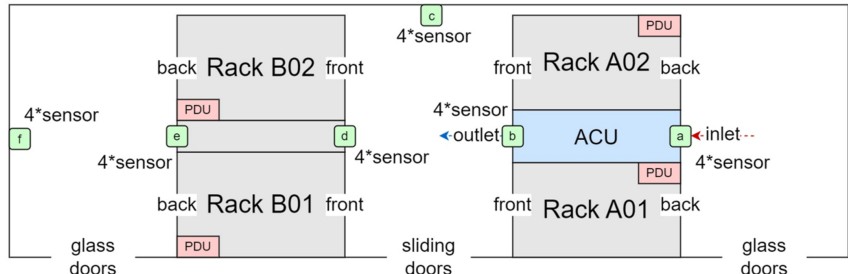

Figure 13: The layout illustration of the real-world DC cooling control testbed environment.

- **"VAE-rep"**: The variational autoencoder (VAE) (Kingma & Welling, 2014) is built based on the "AE-rep" model by introducing additional KL divergence loss terms. Specifically, the encoder outputs parameters of a Gaussian distribution in the latent space, and the latent representations are sampled using the reparameterization trick. The VAE is trained using a combined loss function that includes both the reconstruction loss and the KL divergence loss, which regularizes the latent space to follow a prior distribution. The inverse dynamic module is trained simultaneously with the VAE, sharing the latent space and optimizing for both the reconstruction of the input data and the prediction of actions.
- **"Contras-rep"**: We utilize the NT-Xent loss (Normalized Temperature-Scaled Cross Entropy Loss) used in SimCLR (Chen et al., 2020) within the latent representation space on top of the "AE-rep" model. The overall loss function combines the contrastive loss with the reconstruction loss, ensuring that the latent space not only captures the structure of the data but also learns semantically meaningful representations that are robust to variations. The inverse dynamic module is trained simultaneously within the latent space to predict actions.

### D.2 EXPERIMENT DETAILS OF REAL-WORLD INDUSTRIAL CONTROL TEST ENVIRONMENT.

We adapted the figure from (Zhan et al., 2025) to illustrate the layout structure of the real-world DC cooling control testbed. As shown in the Section D, the testbed comprises 22 server units and an inter-rack air conditioning unit (ACU) positioned between Rack 1 and Rack 2, supplemented by 24 temperature and humidity sensors (organized into six monitoring sets) to capture spatial thermal dynamics within the environment. Notably, the ACU employs compressor-driven cooling, with fan operation and compressor workload constituting the primary sources of energy expenditure. The thermal regulation is achieved by modulating the ACU's entering air temperature (EAT) setpoint to maintain the cold aisle temperature (CAT) below a predefined safety threshold. The energy-saving objective is to improve the energy efficiency of the DC's cooling systems (minimizing the ACLF) while satisfying thermal safety constraints.

We leverage a dataset of 43k real-world operational samples recorded at 2-minute intervals over 61 days with 105 state-action features. During the training process, we utilize the identical reward function and follow the same experimental protocols outlined in (Zhan et al., 2025). To ensure rigorous benchmarking, we adopt the same challenging thermal constraint (set the CAT threshold as 22°C) for comparative evaluation of TELS performance. Following the testing protocol in (Zhan et al., 2025), we ran our RL policy on the testbed continuously for 2 hours, which issues control commands every 2 minutes. We collected and aggregated all the energy-saving measurements at 2-minute intervals to calculate the final ACLF metric.

## E BROADER IMPACT

While training reinforcement learning (RL) agents on large-scale offline datasets has been extensively studied, real-world applications often face prohibitive data scarcity and collection costs. This necessitates offline RL methods that achieve reliable performance in small-sample regimes. To address this challenge, we introduce a highly sample-efficient offline RL algorithm to learn high-performing policies from extremely limited data. We empirically validate its efficacy through deployment on a real-world data center cooling control testbed, establishing its practical viability.

Our approach highlights a promising pathway for advancing sample-efficient offline RL in resource-constrained settings. A potential limitation is the inherent risk of unreliable or unsafe actions within historical datasets, which may mislead policy learning.

## F    LEARNING CURVES

The following are the learning curves of TS-IDM and the T-symmetry regularized guide-policy optimization in TELS on the reduced-size D4RL MuJoCo and Antmaze datasets. We evaluate the policy with 10 episodes over 5 random seeds.

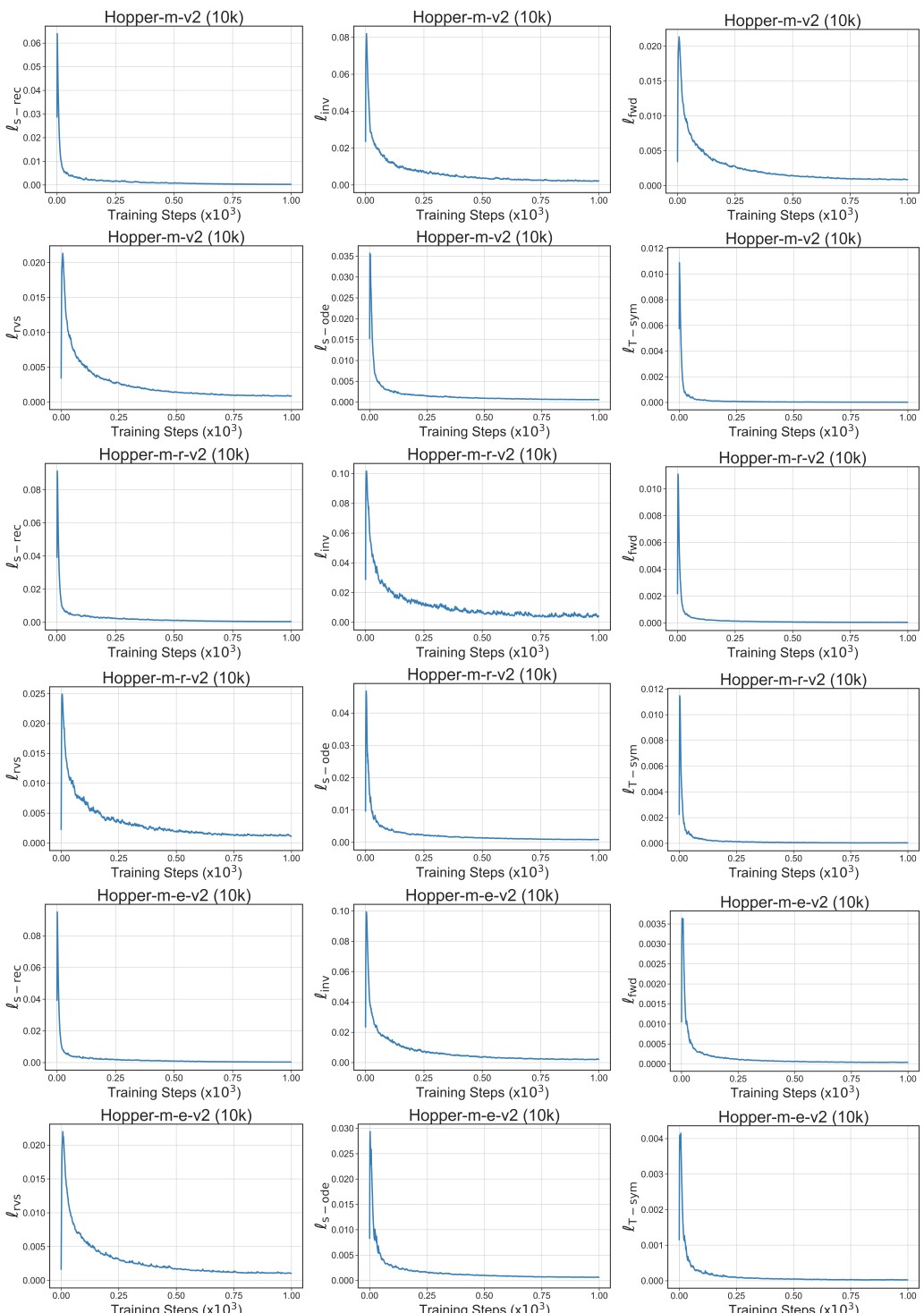

Figure 14: Learning curves of the overall and each individual loss terms in TS-IDM for Hopper tasks.

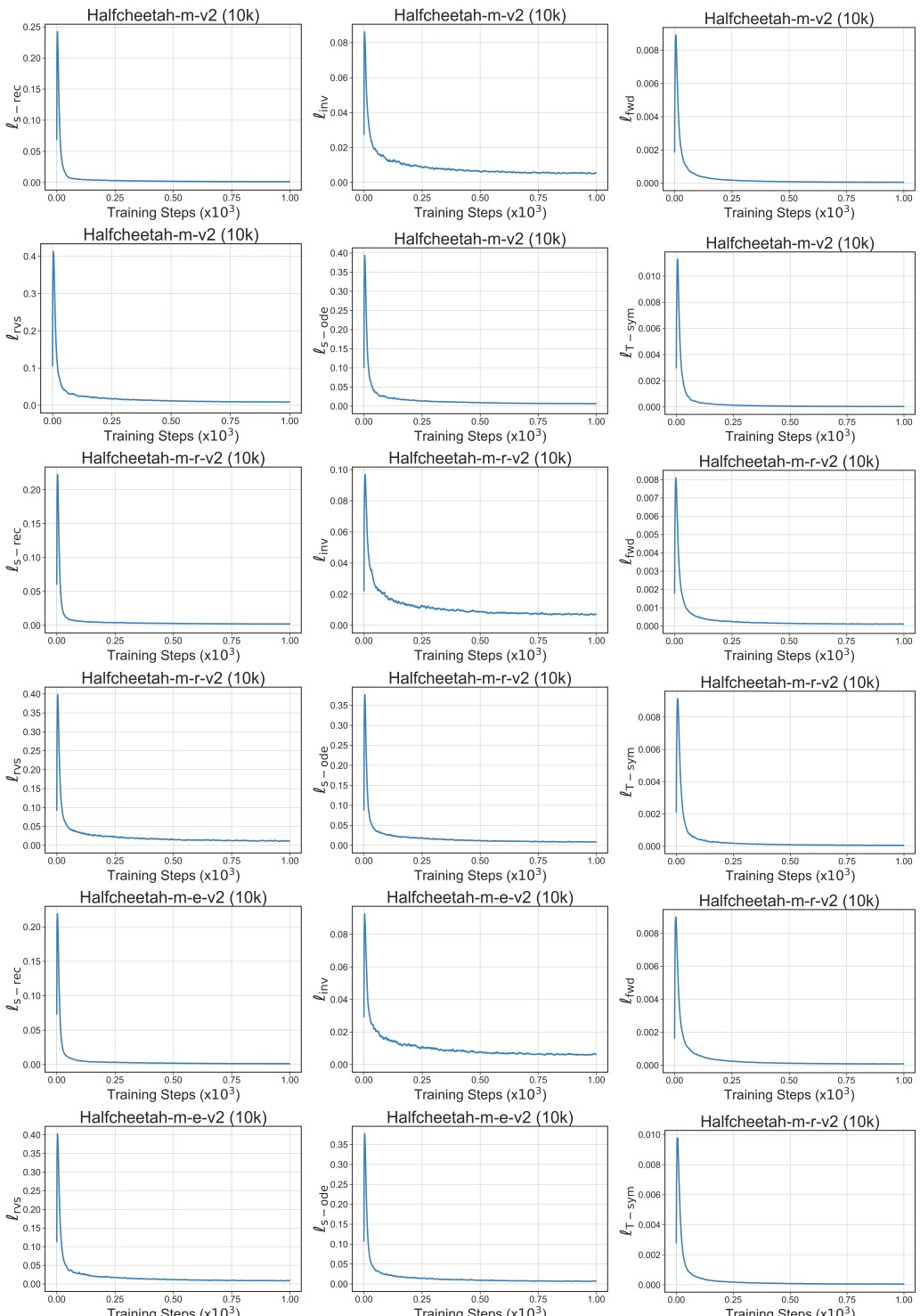

Figure 15: Learning curves of the overall and each individual loss terms in TS-IDM for Halfcheetah tasks.

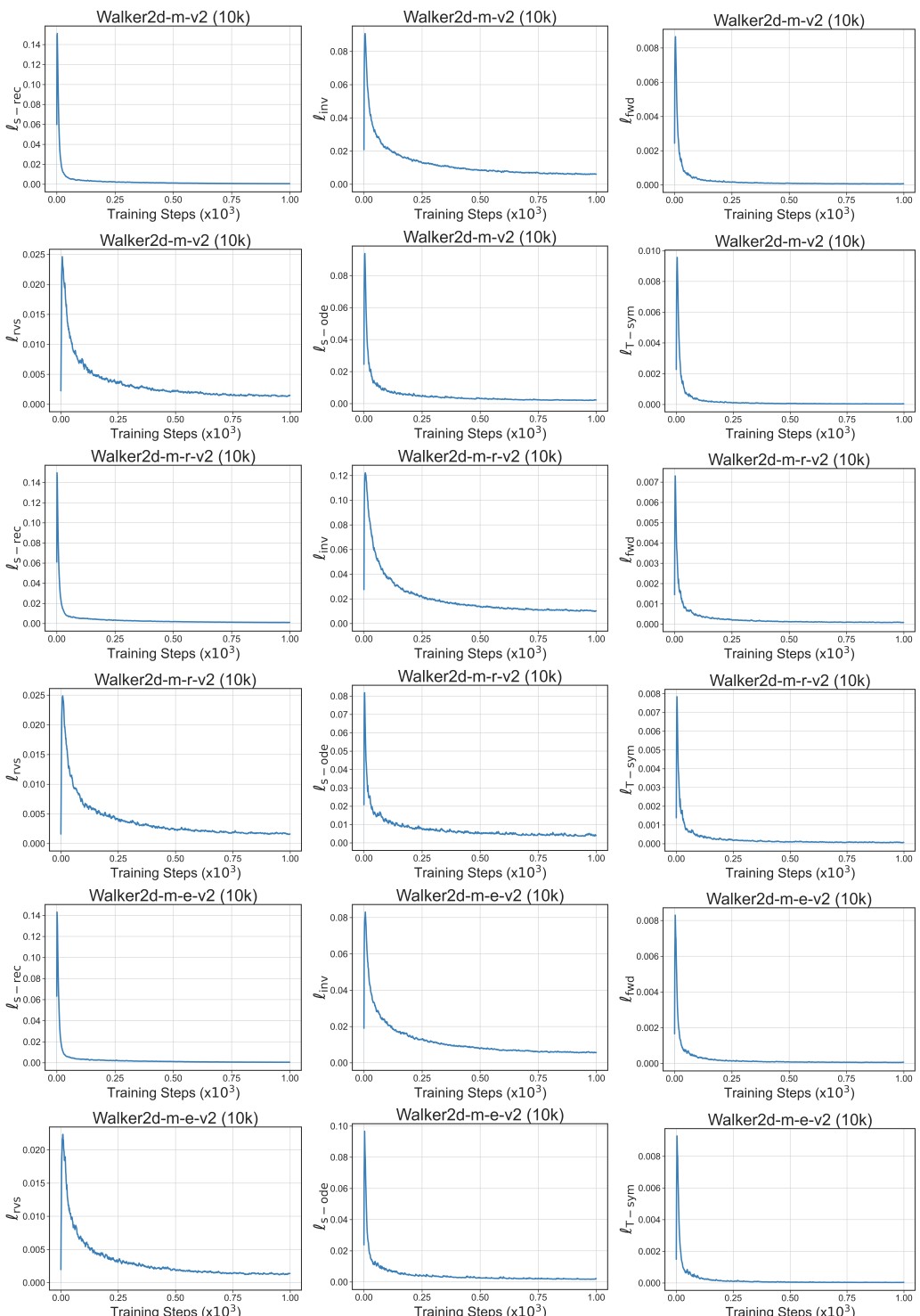

Figure 16: Learning curves of the overall and each individual loss terms in TS-IDM for Walker2d tasks.

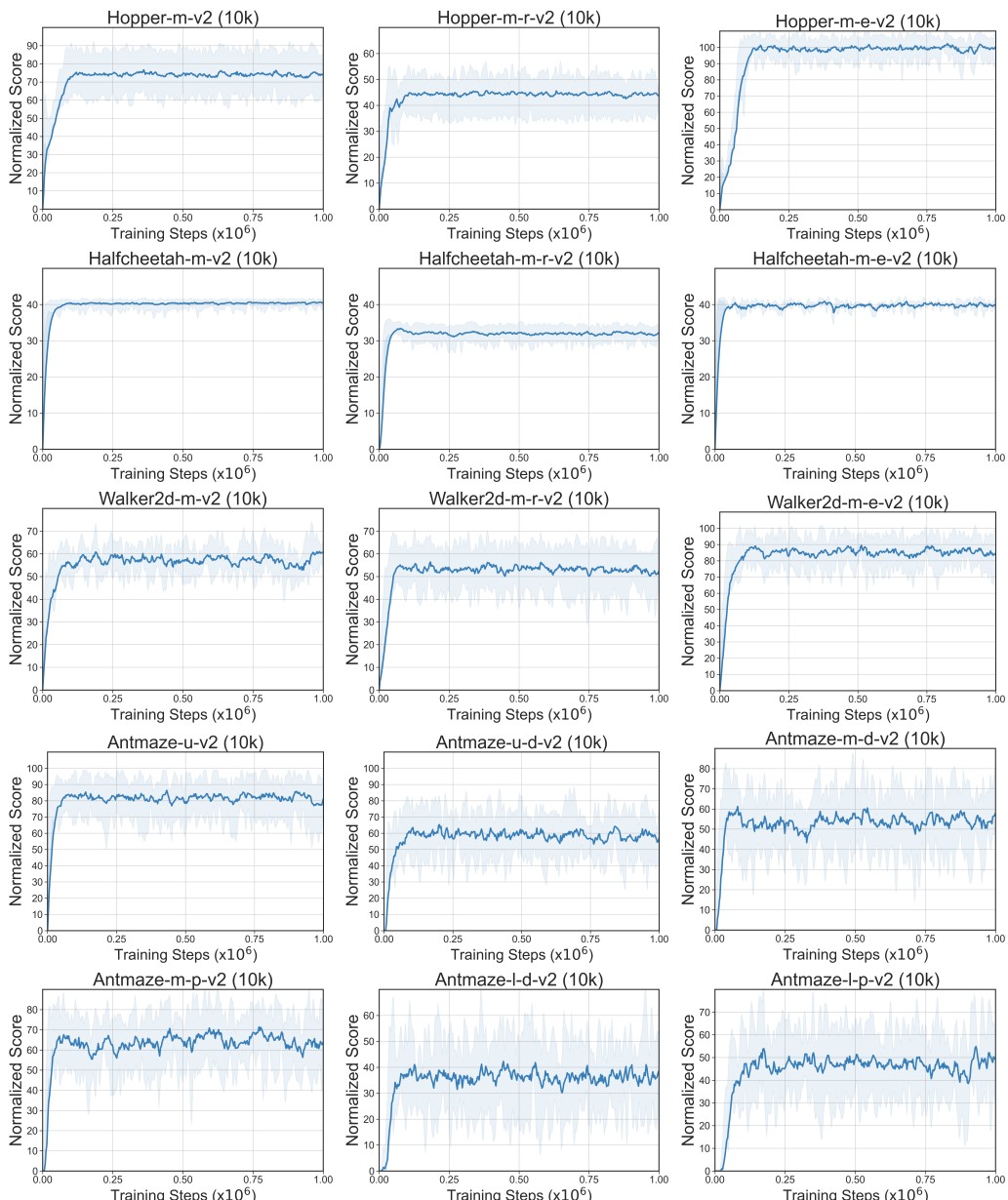

Figure 17: Learning curves of policy optimization in TELS for D4RL MuJoCo and Antmaze tasks with reduced-size datasets. We evaluate the policy within 10 episodes over 5 random seeds.

