# OpenReview forum: "Sample Efficient Offline RL via T-Symmetry Enforced Latent State-Stitching"
_ICLR.cc/2026/Conference — ICLR 2026 Poster_

### Official Review · Reviewer_96gJ · 2025-10-29

**Soundness:** 3
**Presentation:** 2
**Contribution:** 3
**Rating:** 6
**Confidence:** 3

**Summary:**

The paper proposes a sample-efficient offline RL method, TELS. The method builds on a T-symmetry-enforced inverse dynamics model (TS-IDM) for learning a latent space and offline policy optimization in the latent space. A guide policy selects a target for the next state, and the final action is extracted from the latent inverse dynamics module. The method is benchmarked on D4RL in a small-sample setting and on a real-world industrial control environment, and the method is found to outperform existing offline RL methods on these benchmarks.

**Strengths:**

- The method with the latent ODE and symmetry design is well-motivated and connected to prior work. The method appears to be novel and sound.
- The method shows significant sample efficiency improvements over the chosen baselines.
- The ablations in 4.2 are highly interesting, and they clearly illustrate the value of the ODE property enforcement and the T-symmetry consistency loss. The OOD experiments in Figure 3 are likewise impressive, and the method seems to be fast to train.
- The fact that the learned representation also benefits IQL and TD3+BC supports the claim that the latent-space learning is valuable and generalizable.
- Even though the overall dynamic model learning objective (Eq. 7) consists of multiple terms, equal weighting of all the loss terms is nice and speaks to the robustness of the method.

**Weaknesses:**

- No code made available.
- You could consider citing and comparing to existing trajectory stitching methods in offline RL like DiffStitch [1].
- The real-world experiment (Table 2) is very cool, but its significance is somewhat hard to assess. Given that the experiments were run once (as I understand the paper), there are no error bars, and the differences in results could simply be due to, e.g., how well the initial hyperparameter guess worked out.
- The paper claims to "completely bypass the conservatism issue caused by the action-level regularization". However, the deterministic policy loss (Eq. 9) simply replaces the TD3+BC-like action-regularization with a next state-regularization term. The stochastic variant uses AWR over the next states. It is still somewhat unclear if this next state-regularization leads to different behavior and/or learning dynamics than action-level regularization in practice. The empirical results indicate that the difference-maker is the latent space and the advantage-weighed regression for the maze tasks.
- The weight of \alpha (Table 10) differs greatly between environments, raising questions about hyperparameter sensitivity.
- As acknowledged by the authors, the ODE and T-symmetry regularizations can limit the model's expressive power, making this method specifically useful for small datasets.
- Minor: Some issues with presentation & typos, for instance: Equations 9, 10 use h_{ivs} instead of h_{inv}, Appendix E "Border Impact", Zhan et al. 2025a and 2025b is the same paper cited twice, comparision (Table 11 caption), exhibt on L126, Training Perparameters (Table 9).

[1] Li, G., Shan, Y., Zhu, Z., Long, T., & Zhang, W. (2024). Diffstitch: Boosting offline reinforcement learning with diffusion-based trajectory stitching. ICML.

**Questions:**

- Eq3: Do you use stop-gradient at all, e.g. for z_s, z_{s'} in the second term?
- Would it be possible to ablate the second term of the deterministic policy loss by replacing it with a BC-like term in the action space?
- Would you expect this method to be suitable for offline-to-online adaptation? I interpret that the l_{T-sym} term in the policy loss is essentially preventing the policy from going where the dynamics model is inaccurate, and if you fine-tune the encoder, you might run into non-stationarity issues. Is my understanding correct?
- Could you discuss the potential trade-offs of using the T-symmetry prior? It could be a problem in manipulation tasks or other tasks with impacts. On the other hand, based on Table 1, the method works reasonably well in Adroit. Evaluating also on *-cloned and/or *-expert would strengthen the case.
- Would the computation of the Jacobians be a bottleneck in the case of, for example, visual offline RL tasks, where the networks would benefit from being scaled up? Probably manageable, if you use a frozen backbone? If you have time and resources, V-D4RL experiments, for instance, could add value to the paper.

---

> ### Author Response · Authors · 2025-11-24
> **Author Response to Reviewer 96gJ (1/3)**
>
> We thank the reviewer for the constructive comments and positive feedback on our paper. Regarding the reviewer's comments, we provide the following responses.
>
> > **W1. No code made available.**
> * We thank the reviewer for the comment. We will release the PyTorch and Jax version of our code after the paper is accepted.
>
> > **W2. Citing and comparing to existing trajectory stitching methods in offline RL like DiffStitch.**
> * We thank the reviewer for suggesting this related work. To evaluate the effectiveness of DiffStitch under the small-sample setting, we use DiffStitch's official implementation to train its dynamics model and diffusion model, and use them to generate samples then merge with the original dataset. We train TD3+BC and IQL on the resulting data following the same procedure as in the DiffStitch paper. We report the evaluation results in the following table:
>
> **Evaluation results under 10k sample**
>
> |**Task**|**IQL**|**DStitch-IQL**|**TD3+BC**|**DStitch-TD3+BC**|**TELS**|
> |-|-|-|-|-|-|
> |Hopper-m|46.7 $\pm$ 6.5|35.4$\pm$ 10.6|40.1 $\pm$ 18.6|44.1 $\pm$ 8.4|**77.3 $\pm$ 10.7**|
> |Hopper-mr| 13.4 $\pm$ 3.1|12.7$\pm$ 6.6| 7.3 $\pm$ 6.1|20.1 $\pm$ 3.3 |**42.3 $\pm$ 3.5**|
> |Hopper-me|34.3 $\pm$ 8.7|21.8$\pm$ 6.4|17.8 $\pm$ 7.9|32.2 $\pm$ 4.3 |**100.9 $\pm$ 6.8**|
> |Halfcheeta-m|29.9 $\pm$ 0.1 |18.9$\pm$ 1.9| 16.4 $\pm$ 10.2|17.5 $\pm$ 4.9|**40.8 $\pm$ 0.6**|
> |Halfcheeta-mr| 22.7 $\pm$ 6.4 |16.7$\pm$ 3.8|17.9 $\pm$ 9.5|9.8 $\pm$ 3.1|**33.2 $\pm$ 1.0**|
> |Halfcheeta-me| 10.5 $\pm$ 8.8|9.9$\pm$ 1.8| 15.4 $\pm$ 10.7|9.6 $\pm$ 4.0|**40.7 $\pm$ 1.2**|
> |Walker2d-m| 22.5 $\pm$ 3.8|4.0$\pm$ 1.6|7.4 $\pm$ 13.1|10.7$\pm$ 1.3|**62.4 $\pm$ 5.3**|
> |Walker2d-mr| 10.7 $\pm$ 11.9 | 5.8$\pm$ 5.6|5.7 $\pm$ 5.8 |12.7$\pm$ 0.4|**54.8 $\pm$ 6.0**|
> |Walker2d-me|26.5 $\pm$ 8.6|1.7$\pm$ 2.2|7.9 $\pm$ 9.1 |6.7$\pm$ 2.2|**87.4 $\pm$ 13.3**|
>
> * During the experiment with DiffStitch, we observed that it struggles to learn accurate constituent models (forward dynamics, reward, inverse dynamics, and diffusion models) in small dataset settings. It suffers from low-quality generation, most of its generated data do not satisfy the validity thresholds (i.e., dynamic, action, and reward consistency) originally defined in the DiffStitch protocol.
> * To address this, we progressively relaxed the cosine similarity threshold—used to identify potential state pairs for stitching—from 0.9 to 0.7. Even with this relaxation, the generation process remained inefficient, yielding only about 1k synthetic data points in approximately one hour. For comparison, under full dataset settings, DiffStitch typically generates ~2M samples in two hours. This discrepancy highlights the extreme difficulty of generating valid data under small-sample settings.
> * The above empirical evaluation further confirms that due to the poor quality of the synthetic samples, incorporating data generated by DiffStitch yielded negligible performance improvements. In some cases, performance even degraded, likely due to the noise introduced by the compromised synthetic data.
>
> > **W3. The real-world experiment (Table 2) is very cool, but its significance is somewhat hard to assess. There are no error bars, and the differences in results could simply be due to, e.g., how well the initial hyperparameter guess worked out.**
>
> * We thank the reviewer for the recognition of our real-world deployment. We wish to clarify that the results in Table 2 do not represent a "one-shot" snapshot. Following the testing protocol in [1], we ran our RL policy on the testbed continuously for 2 hours, which issues control commands every 2 minutes. We collected and aggregated all the energy-saving measurements at 2-minute intervals to calculate the final ACLF metric. As the reviewer can see, the ACLF is already an aggregated metric reflecting long-term operation. But we agree with the reviewer that conducting more experiments could further strengthen the evaluation. We have added more information about the evaluation process in our revised manuscript and will conduct more experiments in our final paper.

---

> ### Author Response · Authors · 2025-11-24
> **Author Response to Reviewer 96gJ (2/3)**
>
> > **W4&Q2. It is still somewhat unclear if this next state-regularization leads to different behavior and/or learning dynamics than action-level regularization in practice. Would it be possible to ablate the second term of the deterministic policy loss by replacing it with a BC-like term in the action space?**
>
> * We thank the reviewer's in-depth comments. In our paper, we have conducted a series of controlled experiments to demonstrate the effectiveness of our design.
> * For the reviewer mentioned case that replaces the state-level regularization term with BC action constraints, we refer the reviewer to the TSRL baseline. It learns a T-symmetry regularized state-action encoder $\phi(s,a)$ and optimizes a deterministic policy loss with a BC constraint in the action space. As shown in Table 1, we can see that TSRL significantly underperforms our proposed TELS.
> * In Fig. 4(left) of our paper, we also tested the performance of TD3+BC with the state representation $\phi_s(s)$ learned using our proposed TS-IDM. The result shows that our learned latent representation indeed improves TD3+BC's performance, but the final performance still has a significant gap as compared to the scores of TELS in Table 1.
> * Lastly, in Table 4, Appendix A of our paper, we further conducted the experiments by combining the TSRL state-action representation with POR's state-stitching mechanism, the results show that its performance is even worse than TSRL.
> * Therefore, the above controlled experiments show that both the TS-IDM's well-behaved representation and our latent state space optimization scheme matter. Removing either of them will lead to a performance drop.
>
> > **W5. The weight of $\alpha$ (Table 11) differs greatly between environments, raising questions about hyperparameter sensitivity.**
>
> * Note that $\alpha$ mainly impacts reward scaling. AntMaze has a very different reward structure (0-1 sparse reward), whereas MuJoCo has a dense reward function. Note that many other offline RL papers, such as IQL, POR, etc., also use a much larger AWR $\alpha$ value for AntMaze tasks, we refer the reviewer to check the hyperparameter settings in these papers. We'd also like to mention that for all our AntMaze and MuJoCo tasks, we use a consistent set of hyperparameters for all tasks without tuning, whereas many other offline RL papers tune their hyperparameters for each of their tasks to get good results.
>
> > **W6. As acknowledged by the authors, the ODE and T-symmetry regularizations can limit the model's expressive power, making this method specifically useful for small datasets.**
>
> Yes, but as we have shown in Table 5 in Appendix B.1, even under the full dataset setting, the performance of TELS is still not bad. Moreover, as we have discussed in Appendix B.5, reducing the strength of the ODE and T-symmetry regularization (using a smaller $\beta$) will improve the performance of TELS under large datasets, as when the amount of data is abundant, the stringent extra regularization can be reduced.
>
> > **W7. Minor typos.**
>
> We thank the reviewer for this constructive comment. We have corrected all the identified issues in our updated manuscript.
>
> > **Q1. Do you use stop-gradient at all, e.g. for z_s, z_{s'}?**
>
> No, we did not detach the gradient for $z_s$ and $z_{s'}$ in our original implementation. This is because that couples all the components inside TS-IDM, and ensures the inverse dynamics model ($h_{inv}$) is jointly optimized.
> * To further investigate this, we conducted additional ablation studies in the small-sample setting, comparing the standard TELS implementation against a variant where the gradient flow on $z_s$ and $z_{s'}$ is detached ("TELS (detach)"). The results are summarized below:
>
> **Performance comparision under 10k sample**
>
> |**Task**|**TELS(detach)**|**TELS**|
> |-|-|-|
> |Hopper-m|74.7 $\pm$ 4.1 ($\downarrow 3.4$%) |**77.3 $\pm$ 10.7**|
> |Hopper-me| 55.5 $\pm$ 33.5 ($\downarrow 45.0$%) |**100.9 $\pm$ 6.8**|
> |Walker2d-m|47.9 $\pm$  1.7 ($\downarrow 23.4$%) |**62.4 $\pm$ 5.3**|
> |Walker2d-me| 61.8 $\pm$ 11.3 ($\downarrow 29.3$%) |**87.4 $\pm$ 13.3**|
>
> * The results show that detaching the gradient leads to significant performance degradation across all tasks. This indicates that the supervisory signal from the inverse dynamics task is crucial. Without it, the learned state representations may diverge from the structure required for reliable action inference. This phenomenon is particularly critical in mixed-quality datasets (e.g., Hopper-me), where we observe a severe performance decline of approximately 45.0%.

---

> ### Author Response · Authors · 2025-11-24
> **Author Response to Reviewer 96gJ (3/3)**
>
> > **Q3. Could you expect this method to be suitable for offline-to-online adaptation?.**
>
> * In principle, our method can also be used in the offline-to-online setting, as by learning a fundamental dynamics pattern, our proposed TS-IDM learns a data distribution-agnostic representation. During the online phase, we can update both TS-IDM and the policy using the extra online data for better performance. But this setting actually deviates from the motivation of our work, as we mainly focus on how to learn well-behaved representations and policies using very limited data. If unrestricted online data collection is allowed, such strong consistency regularization in TELS is not really necessary for policy learning.
>
> > **Q4.  Could you discuss the potential trade-offs of using the T-symmetry prior. Evaluating also on -cloned and/or -expert would strengthen the case.**
>
> * We have some discussion on the trade-off of using the T-symmetry prior in the "Ablation of $\beta$ in TS-IDM" section of Appendix B.5. In general, we find that smaller datasets benefit from relatively strong T-symmetry and ODE regularization; larger datasets contain sufficient information from data samples, thus requiring less regularization. We refer the reviewer to our paper for a more detailed discussion.
> * Regarding the results on Adroit-cloned/expert tasks, we have conducted additional experiments during rebuttal. As the original Adroit-cloned/expert tasks have much larger datasets (500k) as compared to Adroit-human tasks (5k samples), substantially reducing the learning difficulty, we therefore test our methods against baselines on a more challenging reduced-size setting with 10k samples. The results are presented in the following table, where TELS still achieves strong performance.
>
> |**Task**|Size(ratio)|**BC**|**TD3+BC**|**MOPO**|**COMBO**|**CQL**|**IQL**|**DOGE**|**IDQL**|**POR**|**TSRL**|**TELS**|
> |-|-|-|-|-|-|-|-|-|-|-|-|-|
> |pen-cloned|10k(2%)|37.4 $\pm$ 37.6 | 0.1 $\pm$ 3.0 | 0.1 $\pm$ 0.1 | 0.7 $\pm$ 0.2 | 1.5 $\pm$ 4.8 | 35.6 $\pm$ 30.5 | 30.1 $\pm$ 19.7 | 64.4 $\pm$ 15.1 | 43.6 $\pm$ 5.8 | 41.6 $\pm$ 27.5 | **69.7 $\pm$ 12.6**
> |pen-expert|10k(2%)| 27.6 $\pm$ 21.3 | 5.2 $\pm$ 2.7 | 1.2 $\pm$ 0.3 | 2.5 $\pm$ 0.4 | 3.6 $\pm$ 4.5 | 68.9 $\pm$ 24.3 | 31.1 $\pm$ 19.3 | **104.6 $\pm$ 3.8** | 61.2 $\pm$ 21.0 | 65.6 $\pm$ 22.8 | **105.7 $\pm$ 12.1**
> |hammer-cloned|10k(2%)| 0.3 $\pm$ 0.4 | 0.2 $\pm$  0.1 | 0.1 $\pm$ 0.1| 0.2 $\pm$ 0.1 | 0.2 $\pm$ 0.1 | 0.4 $\pm$ 0.2 | 0.3 $\pm$ 0.1 | 0.8 $\pm$ 0.3 | 0.1 $\pm$ 0.1 | 0.6 $\pm$ 0.3 | 0.6 $\pm$ 0.2 |
> |hammer-expert|10k(2%)| 0.2 $\pm$ 0.1| 0.5 $\pm$ 0.2 | 0.1 $\pm$ 0.1 | 0.2 $\pm$ 0.1|1.2 $\pm$ 1.1 | 70.3 $\pm$ 30.3 | 0.6 $\pm$ 0.3 | **91.7 $\pm$ 12.9** | 2.7 $\pm$ 2.6 | 77.6 $\pm$ 31.2 | **91.5 $\pm$ 25.9**
> |door-cloned|10k(2%)| 0.1 $\pm$ 0.1 | 0.3 $\pm$ 0.1 | 0.2 $\pm$ 0.1| 0.1 $\pm$ 0.3 | 0.2 $\pm$ 0.1| 1.5 $\pm$ 0.8 | 0.5 $\pm$ 0.5| 0.1 $\pm$ 0.1|0.1 $\pm$ 0.1 | 0.1 $\pm$ 0.3 | **7.6 $\pm$ 2.3**
> |door-expert|10k(2%)|1.2 $\pm$  1.1 | 5.2 $\pm$ 3.1 |1.5 $\pm$ 1.2 | 3.5 $\pm$ 1.1 |20.3 $\pm$ 15.7 | 79.2 $\pm$ 8.8 | 0.5 $\pm$ 0.1 |  98.3 $\pm$ 5.5 | 0.7 $\pm$ 0.3 | 46.3 $\pm$ 12.5 | **101.8 $\pm$ 8.5**
> |relocate-cloned|10k(2%)|0.2 $\pm$ 0.1 | 0.3 $\pm$ 0.1 | 0.3 $\pm$ 0.2 | 0.1 $\pm$ 0.1 | 0.3 $\pm$ 0.1 | 0.1 $\pm$ 0.5 |0.1 $\pm$ 0.1| 0.2 $\pm$ 0.2 | 0.1 $\pm$ 0.1 | 0.2 $\pm$ 0.1 | 0.2 $\pm$ 0.1
> |relocate-expert|10k(2%)|0.6 $\pm$  0.1| 0.1 $\pm$ 0.1 | 0.1 $\pm$ 0.2 |1.5 $\pm$ 1.2 | 0.2 $\pm$ 0.1 | 31.1 $\pm$ 8.4 | 0.3 $\pm$ 0.5| **85.5 $\pm$ 12.7** | 0.2 $\pm$ 0.1|45.2 $\pm$ 15.3 | **85.6  $\pm$  12.1** |
>
>
> > **Q5. Would the computation of the Jacobians be a bottleneck? Scaling up to visual input.**
>
> - As we have mentioned in Appendix C.2, we actually directly compute Jacobian-vector product (JVP) when evaluating $\nabla \phi_s(s)\dot{s}$, which can be naturally supported in Jax or torch.vmap() function in PyTorch for efficient computation. Computing the Jacobian-vector product is much cheaper as compared to first computing the Jacobian $\nabla \phi_s(s)$ and then multiplying by the vector $\dot{s}$, hence it will not cause computation issues. In our Jax implementation, training TS-IDM on the 10k Hopper-m datasets only takes 5 minutes.
> - For scaling up to larger tasks with visual inputs, it is suggested to leverage a pre-trained vision encoder and keep it frozen during TS-IDM training. In this way, TS-IDM can learn a new ODE & T-symmetry compatible latent representation on top of the original visual embedding, while keeping the computation cost manageable. We will conduct more experiments in our future work to further scale up TELS for visual control tasks.
>
> **References:**
>
> [1] Zhan, et al. "Data center cooling system optimization using offline reinforcement learning." ICLR, 2025.

---

### Official Review · Reviewer_GJr5 · 2025-10-31

**Soundness:** 3
**Presentation:** 3
**Contribution:** 3
**Rating:** 8
**Confidence:** 4

**Summary:**

This paper proposes TELS, a sample-efficient offline reinforcement learning (RL) algorithm that leverages time-reversal symmetry (T-symmetry) to learn latent state and action representations.
By enforcing T-symmetry in an inverse dynamics model and optimising a guide policy entirely in latent space, the method improves out-of-distribution generalisation and sample efficiency compared to existing offline RL approaches.
The paper is clearly written, well-motivated, and supported by thorough experiments on both D4RL benchmarks and a real-world industrial control task, demonstrating strong empirical gains in small-sample regimes.

**Strengths:**

- Clarity and presentation:

    The paper is clearly written and well-organised. The introduction and preliminaries effectively motivate the work, set the stage for the proposed approach, and position it well in relation to prior research. The method and experimental sections are easy to follow, and the paper maintains a strong logical flow throughout.

- Methodological contribution:

    Although the proposed method is a relatively straightforward extension of existing offline RL techniques, the authors demonstrate that this extension—enforcing time-reversal symmetry in the latent dynamics model—has a substantial positive impact on performance and generalisation.

- Empirical evaluation:

    The experiments are comprehensive and well-designed. The results show strong performance, particularly in data-limited offline RL settings, and provide convincing evidence of the method’s effectiveness and sample efficiency. The inclusion of both D4RL benchmarks and a real-world industrial control environment strengthens the paper’s practical relevance.

**Weaknesses:**

- Captions lacking detail:

    In general, table and figure captions do not provide enough information.

    - Figure 1: The caption should explain how to interpret the figure.

    - Table 1: The caption should clarify what “±” represents (e.g., standard deviation?), what the bolded values indicate (best method overall or under a statistical test such as a paired t-test?), and where readers can find details about the reduced-size datasets.

    - Table 2: In the third row, two values are bolded—please clarify what bolding signifies in the caption.

    - Table 3: The caption should explain what each row represents, as this is not clear on first reading.

- Missing normalisation details:

    The paper does not specify how the normalised scores are computed. The authors should include this information, ideally in the captions of tables or figures where normalised scores are reported.

- Figure 2 presentation:

    Figure 2 could be improved by aggregating results across both environments (and potentially more). The authors may find the rliable package useful for this purpose. I recommend reporting the interquartile mean along with 95% stratified bootstrap confidence intervals.

- Clarification on thermal safety violations (Line 358):

    The paragraph beginning on Line 358 does not make it clear what causes thermal safety violations or what they reveal about the algorithm’s behaviour. Please add a brief sentence providing intuition.

Minor issues:
- Line 78: When introducing “latent representations,” clarify that these refer to latent state and action representations.
- Line 102: The citation should be presented textually (i.e., integrated into the sentence rather than in parentheses).
- Line 311: Clarify the meaning of the tilde (~) symbol. Should this instead read “5k–100k”?

**Questions:**

1. One thing that I think has been overlooked is that this method won't work if the environment's transition dynamics are stochastic. Please can the authors comment on this?
2. What is $h\_{ivs}$ in Equation 9?

---

> ### Author Response · Authors · 2025-11-24
> **Author Response to Reviewer GJr5**
>
> We sincerely appreciate the reviewer for the positive feedback and valuable comments. Regarding the comments of the reviewer, we provide the following responses.
>
> > **W1. Captions lack details.**
> * We really appreciate the reviewer for all the constructive suggestions to further improve the quality of our paper. We apologize for the lack of clarity in several parts of our paper. Due to the page limit of ICLR, we had to remove some details from the main text in our initial submission. We have added some details in our revised manuscript and will further improve our paper in the final version.
>
> > **W2. How are the normalised scores computed?**
> * We follow the same normalization scheme as introduced in the D4RL benchmark [1]. It is computed as $\text{Normalized Score} = 100 \times \frac{\text{policy score} - \text{random score}}{\text{expert score} - \text{random score}}$, which is normalized between 0 and 100, evaluated over 100 episodes. The expert score corresponds to the score of a domain-specific expert predefined in the D4RL environment, and the random score corresponds to an agent taking actions uniformly at random across the action space.
>
> > **W3. Aggregating results across both environments.**
> * We thank the reviewer for this constructive comment. We agree that aggregating results across environments or various settings would offer a clearer justification of the algorithm's general performance. We will compute and report the aggregated results in our final paper.
>
> >**W4. What causes thermal safety violations?**
>
> - The thermal safety violations in our experiment correspond to the cases when the cold aisle temperature of the testbed exceeds the safety operation threshold (i.e., $22^\circ \text{C}$). As we have mentioned in Section D.2, we employed the same experiment setup and reward function definition as in Zhan et el. [2], which adds a safety penalty in reward to prevent thermal safety violations. We refer the reviewer to Section C.2 of Zhan et el. [2] for the detailed function form of the reward penalty.
> - Algorithms having thermal safety violations could due to multiple reasons, such as 1) the algorithm fail to learn reasonable policies (e.g., the case for CQL, low return and large percentatge of safety violations); 2) the algorithm learns sub-optimal policies (cannot effectively suppress safety reward penalty); or 3) the learned policy has robustness or OOD generalization issues. In such cases, improper control commands could be issued and cause overheating in the testbed environment.
>
> > **Q1. One thing that I think has been overlooked is that this method won't work if the environment's transition dynamics are stochastic.**
> * We thank the reviewer's insightful question. We acknowledge that the current TELS framework is more suitable to model deterministic environment, as the T-symmetry property in physics is primarily defined on ODE dynamics. However, as we enforce the T-symmetry in a transformed latent state space rather than the raw observation space. This helps alleviate the impact of observational stochasticity to some extent and ensures T-symmetry is operated in a more favorable latent space. We agree that handling high-variance stochasticity is a critical next step, and we will further explore the capacity of TELS in handling such tasks.
>
> > **Q2. What is $h_{ivs}$ in Equation 9?**
>
> We apologize for this typo. It should be $h_{inv}$. We have corrected this in the revised manuscript.
>
> **References:**
>
> [1] Fu, J., et al. D4RL: Datasets for Deep Data-Driven Reinforcement Learning. arXiv 2020.
>
> [2] Zhan, X., et al. Data Center Cooling System Optimization Using Offline Reinforcement Learning. ICLR 2025.

---

> > ### Comment · Reviewer_GJr5 · 2025-11-27
> >
> > Thanks for addressing some of my comments.
> >
> > With regards to the aggregate results, it would be good to see these before the rebuttal is over, as this is kind of the point of the rebuttal. It is no longer acceptable to just report results for 5 seeds in RL. The field has moved to aggregating results over environments (e.g. using rliable) such that results are statistically significant.
> >
> > With regards to the captions, there still seems to be uninformative captions. For example, in Figure 2, it does not mention what the error bars represent, nor how many seeds were used.
> >
> > I am not sure I agree with this comment:
> > > However, as we enforce the T-symmetry in a transformed latent state space rather than the raw observation space. This helps alleviate the impact of observational stochasticity to some extent and ensures T-symmetry is operated in a more favorable latent space.
> >
> > You have a deterministic encoder/decoder (autoencoder style), so your "observation model" is not directly modelling observation noise. But regardless, my comment was meant to encourage you to add this as a limitation and a point for future work.
> >
> > I will maintain my score of 8, however, I will note that my score feels generous unless the above are addressed.

---

> ### Author Response · Authors · 2025-12-03
> **Author Response to Reviewer GJr5**
>
> We thank the reviewer for engaging in the discussion and for the constructive comments. We have revised our manuscript accordingly.
>
> > **1. Aggregate results over environments through rliable plots.**
>
> We appreciate the reviewer's thoughtful suggestion. We have followed the recommendation and added the aggregate performance scores of TELS and all baselines across all locomotion tasks (10k dataset setting) using rliable plots [1]. These results are now included in Appendix B.4, Figure 8 of the updated manuscript. The plots demonstrate that TELS consistently achieves the highest score with the minimum optimality gap compared to all baselines.
>
> > **2. Uninformative caption of Figure 2 and other figures.**
>
> We appreciate the reviewer's advice regarding the figure captions. We have added more descriptive information to the captions of Figure 2 and other figures throughout the manuscript to make them clearer and more informative.
>
> > **3. The comment was meant to encourage you to add this as a limitation and a point for future work.**
>
> We thank the reviewer for this helpful clarification. We have incorporated a discussion on the potential limitations of our work under stochastic dynamics and explicitly designated this as an important direction for future work in the updated manuscript.
>
> **Reference:**
>
> [1] Agarwal, R., Schwarzer, M., Castro, P. S., Courville, A. C., & Bellemare, M. Deep reinforcement learning at the edge of the statistical precipice. NeurIPS 2021.

---

### Official Review · Reviewer_qR8u · 2025-11-01

**Soundness:** 2
**Presentation:** 2
**Contribution:** 2
**Rating:** 6
**Confidence:** 1

**Summary:**

This work introduces a novel T-symmetry enforced latent state-stitching (TELS) algorithm designed to improve the sample efficiency and OOD generalization in offline RL. While existing offline RL methods requires massive dataset to learn reliable policies, and perform poorly when an action has not been observed in training data, TELS addresses those by focusing on maximizing the utility of limited dataset.

The main technical contribution is the TELS mechanism, which learns a low-dimensional latent state representation. It can help creates a robust set of imagined trajectories, which serves as an implicit form of data augmentation and leads to a more reliable policy.

**Strengths:**

1 The TELS demonstrates superior performance, such as D4RL's medium replay datasets, showing its effectiveness in data-limited settings.

2 The TELS policy is trained to navigate a wider and more reliably state-action space, reducing the conservativeness required by typical OOD regularization techiniques.

3 The ability to stitch data allows the algorithm to handle disparate parts of the state space, making it highly suitable for real-world applications where data is collected sporadically.

**Weaknesses:**

1 TELS algorithms requires a complex model, including both RL components and particularly a separate high-quality latent state model with symmetry regularization.

2 The performance of TELS depends heavily on the learned latent space being an accurate and consistent representation.

3 While latent stitching is meant to improve generalization, if the environment dynamics violate the T-symmetry assumption, the learned policy may rely on imagined transitions that are impossible in practice, which may ultimately japardize its performance.

**Questions:**

1 See Weaknesses.

2 Regarding overall learning objective (7) and Table 7: Selecting same weight $\beta$ for $\ell_{dyn}$ and $\ell_{ode}$ is natural while selecting the same weight for the last term $\ell_{T-sym}$ is not intuitive. a) It seems Table 7 is on Hopper-me rather than 10k Hopper-m (Check Table 6 performance score). b) Provide further evidence for weights $(1,1,5)$, $5,5,5$, $(0.5,0.5,1)$ and $(0.5,0.5,0.5)$ could be helpful to show the rationale for selecting a shared $\beta$. A better illustration would be a performance score figure where x axis is weight of $\ell_{dyn},\ell_{ode}$ , y axis is $\ell_{T-sym}$ , and z axis is the score.

3 Can the latent state encoder learned by TELS transferable? What is the performance by re-using the trained encoder from related tasks?

4 It would be great to demonstrate the idea of state-stitching by toy examples to show successful (or unsuccessful) stitched trajectories projected back into the original observation space. This would help the reader to understand the concept of state-stitching.

---

> ### Author Response · Authors · 2025-11-24
> **Author Response to Reviewer qR8u (1/2)**
>
> We appreciate the reviewer for the constructive comments and positive feedback on our paper.  Regarding the reviewer's comments, we provide the following responses.
>
> > **W1. TELS algorithm requires a complex model.**
>
> We'd like to clarify that our proposed method is not that "complicated" as the reviewer thinks. As we have shown in Table 12 in Appendix C.4, training the TS-IDM model only takes 5 min, and the whole RL training can be finished in 20min in our JAX implementation. The entire learning process of TELS also enjoys fast and stable convergence (see learning curves in Fig. 13-15). In fact, all of the sub-components of the TS-IDM, e.g., encoder, decoder, latent forward, reverse, and inverse models, are simple 2-layer MLPs. It is actually a very small model with only 2.8M parameters, much smaller than the sizes of many recent Transformer or diffusion-based offline RL methods.
>
> > **W2.  The performance of TELS depends heavily on the learned latent space**
>
> We acknowledge that the efficacy of TELS relies on the accuracy and consistency of the learned latent space. In fact, all of our model designs are centered around regulating and enforcing its consistency properties, including:
> 1. Reconstruction loss $\ell_{rec}$ to ensure that the latent representation remains faithful to the original state space.
> 2. ODE enforcement through $\ell_{dyn}$ and $\ell_{ode}$ to ensure latent forward and reverse dynamics are ODE functions, and both the encoder and decoder satisfy the ODE property.
> 3. T-symmetry loss $\ell_{T-sym}$ ensures the entire system satisfy T-symmetry property.
>
> These regularization designs create a strongly coupled and consistent system, ensuring that a well-behaved latent space can be stably learned. Furthermore, as we have shown in Fig.4(Right) of our paper, the learned latent space can be much more effective as compared with other representation learning methods.
>
>
> > **W3. If the environment dynamics violate the T-symmetry assumption...which may ultimately jeopardize its performance.**
>
>  * We thank the reviewer for this comment. As discussed in our paper, we adopted an extended version of T-symmetry for generic discrete-time MDP settings following the TSRL paper (i.e.,$F(s,a)=\dot{s}=-G(s',a))$, which can be viewed as an relaxed version of the original definition of T-symmetry in physics: $d\Gamma(x)/dt=-F(\Gamma(x))$, where $\Gamma$ is the time reversal transformation. Note that if we strictly follow the original T-symmetry definition, we actually need to enforce $F(s,a)=\dot{s}=-\widetilde{G}(s',a')$, where the reverse dynamics $\widetilde{G}$ is a ODE function of next state and action $(s',a')$. However, this will make the system easily suffer from non-invertible next actions. By contrast, modeling the reverse dynamics to be $G(s',a)$ allows the forward and reverse dynamics to be conditioned on the same current action $a$, making it far easier to model, as it bypasses the issue of non-invertible next action.
> * Moreover, we specifically enforce the T-symmetry in a transformed latent state space rather than the raw observation space. This helps alleviate the impact of non-deterministic noises and non-invertible dynamics for dynamical systems with cases violating the T-symmetry property, ensuring T-symmetry is enforced in a well-behaved latent space. Similar approaches have also been widely adopted in the control community (such as Koopman theory [1,2] and Sindy [3,4]), which first map the original state-action space into a well-behaved latent space and then construct latent first-order ODE dynamics by fitting the data.

---

> ### Author Response · Authors · 2025-11-24
> **Author Response to Reviewer qR8u (2/2)**
>
> >**Q1. Questions regarding weight $\beta$ selection for $\ell_{dyn}$, $\ell_{ode}$, and $\ell_{T-sym}$.**
>
> * We thank the reviewer for this constructive comment. Yes, Table 8 is on "Hopper-me" rather than "Hopper-m", we apologize for the typo in our original submission. We have corrected it in our revised version. In the tables below, we also conducted the detailed ablations for both "Hopper-m" and "Hopper-me" following the $\beta$ weights combinations suggested by the reviewer.
> * We also agree with the reviewer that $\ell_{T-sym}$ could have a different weight. In our initial paper, we chose the same $\beta$ weight mainly to remove unnecessary hyperparameters and make the algorithm simple to use. Empirically, we found that simply using the same $\beta$ weight for all three loss terms can already achieve good performance. But the reviewer is correct that using a different $\beta$ for $\ell_{T-sym}$ sometimes could achieve even better performance. For example, in the following small-sample "Hopper-m" experiments, making T-symmetry regularization stronger, i.e., using weights $(1, 1, 5)$, actually achieves even better performance as compared to the original scores reported in our paper.
>
> **Hopper-medium-v2 (10k sample)**
> | weight of $\ell_{\text{dyn}}$ | weight of $\ell_{\text{ode}}$| weight of $\ell_{\text{T-sym}}$| Results |
> |:------|:------|:------|:-----|
> | 1     | 1     |  1    | 77.3$\pm$ 10.7|
> | 1     | 1     |  5    |**85.7$\pm$ 11.2**|
> | 5     | 5     | 5     |74.0$\pm$ 8.6 |
> | 5     | 5     | 1     |54.9$\pm$ 6.9 |
> | 0.5   | 0.5   | 1     |77.2$\pm$ 9.6 |
> | 0.5   | 0.5   | 0.5   |74.7$\pm$ 8.8 |
>
> **Hopper-medium-expert-v2 (10k sample)**
> | weight of $\ell_{\text{dyn}}$ | weight of $\ell_{\text{ode}}$| weight of $\ell_{\text{T-sym}}$| Results |
> |:------|:------|:------|:-----|
> | 1     | 1     |  1    |**100.9$\pm$ 6.8**|
> | 1     | 1     |  5    |100.1$\pm$ 0.5 |
> | 5     | 5     | 5     |83.2$\pm$ 11.3 |
> | 5     | 5     | 1     |27.1$\pm$ 16.1 |
> | 0.5   | 0.5   | 1     |90.8$\pm$ 39.6 |
> | 0.5   | 0.5   | 0.5   |71.4$\pm$ 28.3 |
>
> > **Q2. Can the latent state encoder learned by TELS transferable?**
>
> * As long as the task environment has the same underlying dynamics, the learned state encoder should have reasonably good transferability. This is because TS-IDM is designed to learn fundamental, data distribution-agnostic dynamics from the system, hence it could work well in the multi-task data sharing setting as in [5]. However, for problems with shifting or non-stationary dynamics, our proposed method might not work, as it is hard to learn a single fundamental dynamics pattern to account for changing environment dynamics. Nevertheless, we thank the reviewer for this comment and will conduct further exploration to further test our method in cross-task/cross-dynamics settings.
>
> > **Q3. It would be great to demonstrate the idea of state-stitching by toy examples to show successful (or unsuccessful) stitched trajectories projected back into the original observation space.**
>
> * We thank the reviewer for the suggestion. To provide a more intuitive illustration of the learned latent space, we plotted the original data trajectories of the  Hopper-m 10k task, as well as the rollout trajectories of learned TELS and IQL policies on both the original state space and the latent state space (encoded using our TS-IDM state encoder) for comparison. The t-SNE visualization results are presented in Appendix B.6, Figure 12 of our revised paper. We can observe that the learned latent space is much more compact and well-behaved. The policy rollout trajectories form clear, continuous line patterns in our learned latent space, but can be quite noisy in the original state space. Please refer to the updated content of our manuscript for more details.
>
> **References:**
>
> [1] Weissenbacher, M., et al., Koopman q-learning: Offline reinforcement learning via symmetries of dynamics. ICML 2022.
>
> [2] Mezic, I. Spectral properties of dynamical systems, model reduction and decompositions. Nonlinear Dynamics, 2005.
>
> [3] Brunton, S., et al., Discovering governing equations from data by sparse identification of nonlinear dynamical systems. PNAS, 2016.
>
> [4] Champion, K., et al., Data-driven discovery of coordinates and governing equations. PNAS, 2019.
>
> [5] Yu, T., et al. Conservative Data Sharing for Multi-Task Offline Reinforcement Learning. NeurIPS 2021.

---

### Official Review · Reviewer_AstW · 2025-11-01

**Soundness:** 2
**Presentation:** 3
**Contribution:** 1
**Rating:** 4
**Confidence:** 4

**Summary:**

This paper introduces a T-symmetry enforced inverse dynamics model (TS-IDM) that can learn well-behaved and OOD generalizable latent representations, and facilitate action inference. The proposed method outperforme existing offline RL algorithms on small datasets.

**Strengths:**

The paper is clearly written and logically structured.
The proposed method outperforme some offline RL algorithms on small datasets.

**Weaknesses:**

1.	The paper appears to be a straightforward combination of the POR and TSRL.
2.	The compared baselines are relatively outdated and do not include comparisons with some recent and important baseline.

**Questions:**

see weakness

---

> ### Author Response · Authors · 2025-11-24
> **Author Response to Reviewer AstW (1/2)**
>
> We thank the reviewer for the time and effort during the review process.  Regarding the reviewer's comments, we provide the following responses.
>
> > **W1. The paper appears to be a straightforward combination of the POR and TSRL.**
>
> * Simply combining TSRL and POR actually **does NOT work**. In Table 4, Appendix A in our paper (also presented below), we specifically tested the baseline TSRL+POR that directly combines TSRL and POR, i.e., uses the T-symmetry enforced dynamics model (TDM) encoder in TSRL and combines with the policy optimization procedure in POR. As clearly shown in Table 4, in many cases, TSRL+POR is even worse than the original TSRL and POR. This is because the TDM designed in TSRL itself is not compatible with the state-stitching scheme in POR. In fact, TELS introduces lots of designs to fully leverage the benefit of the T-symmetry property and state-stitching scheme. This is also reflected in the huge performance gain of TELS over TSRL, POR, and TSRL+POR.
>
> **Table 4: Performance comparison between TELS, TSRL, POR, and TSRL+POR on reduced-size D4RL datasets.**
> | Task |TELS|TSRL|POR|TSRL+POR |
> | --------- |--------- |--------- | --------- |---------|
> | Hopper-m         |**77.3$\pm$ 10.7**|62.0$\pm$ 3.7|46.4$\pm$ 1.7| 38.5$\pm$ 2.4 |
> | Hopper-mr        |**43.2$\pm$ 3.5**|21.8$\pm$ 8.2|17.4$\pm$ 6.2 | 25.9$\pm$ 5.9 |
> | Hopper-me        |**100.9$\pm$ 6.8**|50.9$\pm$ 8.6|37.9$\pm$ 6.1| 30.3$\pm$ 9.7 |
> | Halfcheetah-m    |**40.8$\pm$ 0.6**|38.4$\pm$ 3.1|33.3$\pm$ 3.2 | 35.2$\pm$ 7.5 |
> | Halfcheetah-mr   |**33.2$\pm$ 1.0**|28.1$\pm$ 3.5|27.5$\pm$ 3.6 | 28.3$\pm$ 4.2 |
> | Halfcheetah-me   |**40.7$\pm$ 1.2**|39.9$\pm$ 21.1|34.7$\pm$ 2.6|38.9 $\pm$ 1.6 |
> | Walker2d-m       |**62.4$\pm$ 5.3**|49.7$\pm$ 10.6|22.2$\pm$ 3.6|25.7$\pm$ 16.9 |
> | Walker2d-mr      |**54.8$\pm$ 6.0**|26.0$\pm$ 11.3|14.8$\pm$ 4.2| 12.9$\pm$ 3.2 |
> | Walker2d-me      |**87.4$\pm$ 13.3**|46.4$\pm$ 17.4|20.1$\pm$ 8.6| 23.8$\pm$ 9.8|
> | Antmaze-u        |**88.7$\pm$ 7.7**|76.1$\pm$ 15.6|42.1$\pm$ 14.2|40.4$\pm$ 18.1|
> | Antmaze-u-d      |**60.9$\pm$ 16.9**|52.2$\pm$ 22.1|6.1$\pm$ 7.3| 6.7$\pm$ 3.1  |
> | Antmaze-m-d      | **47.2$\pm$ 17.3**|0.0|0.0| 0.0|
> | Antmaze-m-p      | **62.9$\pm$ 17.8**|0.0|0.0| 0.0|
> | Antmaze-l-d      | **39.8$\pm$ 14.1**|0.0|0.0| 0.0|
> | Antmaze-l-p      | **47.3$\pm$ 13.1**|0.0|0.0| 0.0|
>
> * We also want to emphasize that our proposed method has many different designs from TSRL and POR, and have addressed many of their drawbacks (see Appendix A or our paper for more detailed discussion). Specifically,
>     - TELS resolved several major flaws/limitations in TSRL, including:
>          1. Switching from state-action representation learning to state representation learning only, avoiding behavior bias in the offline dataset (e.g., dataset-specific action patterns).
>         2. Much better ODE property satisfaction by regulating both the state encoder and decoder. By contrast, TSRL only enforces ODE property in its encoder, leading to learning instability.
>          3. Deviating from the TD3+BC style policy learning to policy optimization in the state space, avoiding conservative action-level constraints and promoting OOD generalization via latent state-stitching.
>          4. Re-usable latent inverse dynamics model for policy extraction. By contrast, TSRL only uses its  T-symmetry enforced dynamics model (TDM) for representation learning, wasting the valuable dynamics information during policy extraction.
>      - TELS also fixed the major drawbacks of POR that require sufficient state-action coverage and lack of learning robustness. TELS achieves this by:
>          1. Perform policy optimization in the well-behaved and generalizable latent state space.
>          2. Incorporating T-symmetry regularization to greatly enhance the OOD robustness and quality of the learned guide-policy $\pi_g$, leading to greatly enhanced performance.

---

> ### Author Response · Authors · 2025-11-24
> **Author Response to Reviewer AstW (2/2)**
>
> > **W2. The compared baselines are relatively outdated and do not include comparisons with some recent and important baseline**.
>
> - We thank the reviewer for this comment. Following suggestion from the reviewer, we have added two baselines ReBRAC [1] and the recent flow Q-learning (FQL) [2] in our main reduced-size D4RL experiments, which are popular recent methods based on Gaussian policy and flow policy respectively. The new results are presented in the following table. As shown in the table below, these two methods again performs poorly under small-sample setting, and have significant gap compared with TELS.
> - Also, as we have discussed in Section 5 Related Work section, many recent offline RL methods that leverage heavy expressive model architectures such as Transformers and diffusion models require extensive amount of data to learn, making them hardly usable for the small-sample setting.
>
> | Task |ReBRAC|FQL|TELS|
> | --------- |--------- |--------- | --------- |
> | Hopper-m         |51.1$\pm$ 6.5|12.1$\pm$ 5.7|**77.3$\pm$ 10.7**|
> | Hopper-mr        |13.7$\pm$ 2.5|17.1$\pm$ 4.5|**43.2$\pm$ 3.5** |
> | Hopper-me        |35.2$\pm$ 6.5|27.9$\pm$ 6.1|**100.9$\pm$ 6.8** |
> | Halfcheetah-m    |32.9$\pm$ 5.6|29.3$\pm$ 4.2 |**40.8$\pm$ 0.6** |
> | Halfcheetah-mr   |24.6$\pm$ 5.7|25.5$\pm$ 5.1 |**33.2$\pm$ 1.0** |
> | Halfcheetah-me   |31.7$\pm$ 4.1|16.7$\pm$ 1.6|**40.7$\pm$ 1.2** |
> | Walker2d-m       |11.7$\pm$ 3.6|13.2$\pm$ 3.4|**62.4$\pm$ 5.3** |
> | Walker2d-mr      |8.1$\pm$ 2.3|5.1$\pm$ 4.2|**54.8$\pm$ 6.0** |
> | Walker2d-me      |15.1$\pm$ 5.4|19.1$\pm$ 4.6|**87.4$\pm$ 13.3**|
> | Antmaze-u        |61.5$\pm$ 12.6|63.3$\pm$ 14.2|**88.7$\pm$ 7.7**|
> | Antmaze-u-d      |46.8$\pm$ 17.3|15.4$\pm$ 2.8|**60.9$\pm$ 16.9**  |
> | Antmaze-m-d      | 0.9 $\pm$ 0.2 | 0.6 $\pm$ 0.3 |  **47.2$\pm$ 17.3**|
> | Antmaze-m-p      |0.3 $\pm$ 0.3 |9.1 $\pm$ 7.7| **62.9$\pm$ 17.8**|
> | Antmaze-l-d      | 0.1 $\pm$ 0.2 | 3.4 $\pm$ 3.5 |  **39.8$\pm$ 14.1**|
> | Antmaze-l-p      |0.3 $\pm$ 0.2| 3.2 $\pm$ 2.1 |  **47.3$\pm$ 13.1**|
>
> **References:**
>
> [1] Tarasov, D., Kurenkov, V., Nikulin, A., & Kolesnikov, S. Revisiting the minimalist approach to offline reinforcement learning. NeurIPS 2023.
>
> [2] Park, S., Li, Q., & Levine, S. Flow q-learning. ICML 2025.

---

### Official Review · Reviewer_Kxst · 2025-11-02

**Soundness:** 3
**Presentation:** 3
**Contribution:** 3
**Rating:** 6
**Confidence:** 3

**Summary:**

The paper proposes TELS, an offline RL algorithm addressing poor sample efficiency and limited OOD generalization in existing methods. It leverages time-reversal symmetry (T-symmetry) and latent state-stitching via a T-symmetry Enforced Inverse Dynamics Model (TS-IDM), which learns generalizable latent representations by enforcing ODE properties. TELS optimizes a latent guide-policy to avoid action-level conservatism and extracts actions via TS-IDM. Experiments on reduced-size D4RL benchmarks and a real-world data center testbed confirm its SOTA small-sample performance.

**Strengths:**

1. Novel and Principled Design: Integrating T-symmetry (a fundamental physical property) and latent state-stitching is clever. The ODE perspective—regularizing both encoders and decoders (unlike TSRL’s encoder-only focus)—ensures distribution-agnostic dynamics capture, while separating latent guide-policy optimization from action extraction avoids over-conservatism, making the method theoretically sound.
2. Comprehensive Experiments: TELS is rigorously tested on 5k–100k sample D4RL tasks (0.5–10% of original sizes) and outperforms 11 baselines. It also works in real-world data center cooling (20.17% ACLF, no thermal violations). Ablations (TS-IDM components, T-symmetry regularization) and OOD tests (Antmaze critical sample deletion) validate robustness.
3. Clear Computational Efficiency: The appendix addresses ODE-related overhead concerns, showing TELS (2-layer MLPs, compact latent space) is efficient—JAX implementation trains in 20 minutes on 10k samples, outpacing baselines like TSRL (160 mins) and CQL (780 mins).

**Weaknesses:**

1. Unaddressed Dynamics Assumptions: TELS assumes invertible, deterministic environment dynamics (required for T-symmetry and inverse modeling), but real-world settings often have non-invertible (e.g., irreversible heat dissipation) or non-deterministic (e.g., noisy sensors) dynamics. No discussion of how TELS performs here limits its applicability.
2. Limited Latent Interpretability: While TS-IDM learns “well-behaved” latent representations, there is no qualitative analysis (e.g., dimensionality reduction, correlation with physical variables) to show what these representations capture, weakening claims about their structure driving generalization.

**Questions:**

TELS uses reverse/backward dynamics for T-symmetry and representation learning, similar to PlayVirtual (Yu et al., NeurIPS 2021), which leverages backward dynamics for RL generalization.  Could the authors clarify the similarities and differences between TELS and PlayVirtual?

---

> ### Author Response · Authors · 2025-11-24
> **Author Response to Reviewer Kxst (1/2)**
>
> We thank the reviewer for the constructive comments and positive feedback on our paper.  Regarding the reviewer's comments, we provide the following responses.
>
> > **W1. TELS assumes invertible, deterministic environment dynamics (required for T-symmetry and inverse modeling), but real-world settings often have non-invertible.**
>  * We thank the reviewer for this insightful comment. As discussed in our paper, we adopted an extended version of T-symmetry for generic discrete-time MDP settings following the TSRL paper (i.e.,$F(s,a)=\dot{s}=-G(s',a))$, which can be viewed as an relaxed version of the original definition of T-symmetry in physics: $d\Gamma(x)/dt=-F(\Gamma(x))$, where $\Gamma$ is the time reversal transformation. Note that if we strictly follow the original T-symmetry definition, we actually need to enforce $F(s,a)=\dot{s}=-\widetilde{G}(s',a')$, where the reverse dynamics $\widetilde{G}$ is a ODE function of next state and action $(s',a')$. However, this will make the system easily suffer from non-invertible next actions. By contrast, modeling the reverse dynamics to be $G(s',a)$ allows the forward and reverse dynamics to be conditioned on the same current action $a$, making it far more easier to model, as it bypasses the issue of non-invertible next action.
>  * Moreover, we specifically enforce the T-symmetry in a transformed latent state space rather than the raw observation space. This helps alleviate the impact of non-deterministic noises and non-invertible dynamics for dynamical systems with cases violating the T-symmetry property, ensuring T-symmetry is enforced in a well-behaved latent space. Similar approaches have also been widely adopted in the control community (such as Koopman theory [1,2] and Sindy [3,4]), which first map the original state-action space into a well-behaved latent space and then construct latent first-order ODE dynamics by fitting the data.
>
> > **W2. Limited Latent Interpretability.**
> * To illustrate the learned representation compactness, we plot the learned representations of the rollout trajectory in the Hopper-m 10k task, as presented in Appendix B.6 Figure 12. We can observe that the latent space is more compact and with a much lower Euclidean distance compared to the original state space, which supports TELS' strong generalizability in the latent space. Please refer to the updated content of our manuscript for more details.

---

> ### Author Response · Authors · 2025-11-24
> **Author Response to Reviewer Kxst (2/2)**
>
> > **Q1. Comparison with PlayVirtual.**
>
> We thank the reviewer for suggesting this related reference and have cited it in our revised paper. PlayVirtual is essentially a model-based approach that employs a bi-directional model to generate virtual trajectories, thereby improving data efficiency. While we discuss the detailed comparison with model-based approaches in Appendix A, we provide a specific comparison with PlayVirtual as follows:
> - Similarities between TELS and PlayVirtual:
>      * Both methods deploy forward and backward dynamics models (e.g., the reverse dynamics model in our case), which aim to predict the future state and previous state, respectively.
>      * Both methods utilize some form of consistency loss to enforce structure within the learned latent space.
> - Differences between TELS and PlayVirtual:
>      * **Primary Objective**: PlayVirtual uses the bi-directional dynamics model to generate synthetic rollouts to enhance performance. Whereas our proposed TS-IDM is primarily designed for state representation learning and action extraction via inverse dynamics, rather than for data generation.
>      * **Learning Schemes**: PlayVirtual models the forward and backward dynamics as conventional prediction models. However, our proposed TS-IDM learns the forward and reverse dynamics as ODE functions. The supervision signals are completely different. For example, in TS-IDM, the supervision signal for latent forward and revserse dynamics is the time-derivative of latent state, e.g., $\nabla_s \phi_s(s)\dot{s}$ and $\nabla_s \phi_s(s')(-\dot{s})$, instead of the $z_{t+1}$ and $z_{t-1}$ in typical prediction models.
>      * **Role of Consistency:** The underlying design logic behind the consistency loss differs a lot. TS-IDM aims to extract fundamental, distribution-agnostic dynamics patterns through T-symmetry and ODE consistency enforcement. In contrast, PlayVirtual uses cycle consistency to guarantee the reliability of the generated data, a process that relies heavily on the prediction accuracy of the dynamics model.
>      * **Small-Sample Effectiveness:** As illustrated in Table 1, in small-sample settings, limited data samples are insufficient for model-based approaches to learn a reliable model for rollout generation. This results in high approximation errors during model rollouts and leads to inferior performance. By contrast, we adopt a representation learning approach, which avoids problematic rollout generation while still preserving useful dynamics-aware information for policy learning.
>
> **References:**
>
> [1] Weissenbacher, M., et al., Koopman qlearning: Offline reinforcement learning via symmetries of dynamics. ICML 2022.
>
> [2] Mezic, I. Spectral properties of dynamical systems, model reduction and decompositions. Nonlinear Dynamics, 2005.
>
> [3] Brunton, S., et al., Discovering governing equations from data by sparse identification of nonlinear dynamical systems. PNAS, 2016.
>
> [4] Champion, K., et al., Data-driven discovery of coordinates and governing equations. PNAS, 2019.

---

### Author Response · Authors · 2025-11-24
**General Response**

We sincerely thank all reviewers for their thorough reading of our manuscript and for providing insightful and constructive feedback. We have revised our paper accordingly (changes are highlighted in blue) and conducted new experiments to address the raised concerns.

The core alterations and additions made in response to the reviewers' valuable suggestions are summarized below:

1.  (For Reviewer Kxst and qR8u): To address the interpretability of the learned representation concerns, we have added Figure 12 in Appendix B.6, which provides a clear visualization of the learned representations by TELS.

2.  (For Reviewer AstW): We have included and discussed comparison results against ReBRAC and FQL in our responses.

3.  (For Reviewer qR8u): We have conducted further experiments on the $\beta$ weight, and present the results and analysis in the responses.

4.  (For Reviewer GJr5): We have incorporated the suggested content modifications to enhance the clarity and flow of the manuscript. We are committed to further polishing the presentation in the final version. We added aggregate performance scores across all locomotion tasks (10k dataset) using rliable plots in Figure 8, Appendix B.4.

5. (For Reviewer 96gJ): We conducted a comparative analysis with the DiffStitching method, presented in our responses, and provided the evaluation results for the Adroit-cloned/expert tasks, which have been added to Table 6 in Appendix B.2.

---

### Author Response · Authors · 2025-12-03
**Summary**

We thank the reviewers for their constructive and valuable feedback. Following the discussion phase, we have substantially revised the paper with new results and ablation studies. As further interaction with the reviewers is no longer possible, we provide this summary to assist the AC in interpreting the current reviews in light of these revisions.

***

**Reviewer Kxst**​ acknowledges our work's novel and principled design, validated by comprehensive experiments and high efficiency.

> **Main concerns**
* Real-world non-invertible dynamics may violate T-symmetry assumptions.
* Limited latent interpretability.
* Comparison with PlayVirtual.


> **Addressed by**
* We clarified in our response that we adopted an extended version of T-symmetry for generic discrete-time MDP settings. By enforcing this in the latent state space rather than the raw observation space, we effectively handle non-invertible dynamics and noise.
* To illustrate the compactness of the learned representations, we added Figure 12 in Appendix B.6, which provides a clear visualization of the representations learned by TELS.
* We provided a specific comparative analysis with PlayVirtual in our corresponding response.

***

**Reviewer AstW** raised concerns regarding the novelty and baseline selection.

> **Main concerns**
* TELS appears to be a straightforward combination of POR and TSRL.
* Baselines are relatively outdated.

> **Addressed by**
* We explicitly emphasized in Table 4, Appendix A that simply combining TSRL and POR does not work. We highlighted our specific design choices that distinguish TELS from these methods and addressed their respective drawbacks.
* We added ReBRAC and Flow Q-learning (FQL) as baselines in our D4RL experiments. The results confirm that TELS maintains a significant performance gap over these recent methods, particularly in small-sample settings.

***

**Reviewer qR8u** recognizes the superior performance and rational structural design of TELS.

> **Main concerns**
* TELS appears requires a complex model.
* What if the enviromentt dynamics violate the T-symetry assumption.
* Demonstrate the learned latent space to demonstrate the state-stitching mehcanism.

> **Addressed by**
* We demonstrated in Table 12 (Appendix C.4) that training the TS-IDM takes only 5 minutes, and the entire RL training finishes in 20 minutes (using our JAX implementation). Furthermore, our sub-components are simple 2-layer MLPs, avoiding the complexity of recent Transformer or diffusion-based methods.
* We clarified in our response that we adopted an extended version of T-symmetry for generic discrete-time MDP settings. By enforcing the T-symmetry in a transformed latent state space rather than the raw observation space, we effectively handle non-invertible dynamics and noise.
* We included Figure 12 in Appendix B.6 to visualize the learned representations and demonstrate the effectiveness of the latent space.


***

**Reviewer GJr5** offers positive feedback and valuable suggestions for improving the clarity of our paper.

> **Main concerns**
* Uninformative figure/table captions.
* Request for aggregate results for better statistical evaluation.
* Discussion on stochastic dynamics.

> **Addressed by**
* We updated the captions for figures and tables throughout the manuscript to provide more context as suggested.
* We added aggregate performance scores across all locomotion tasks (10k dataset) using rliable plots in Figure 8, Appendix B.4.
* We incorporated a discussion on the limitations regarding stochastic dynamics and explicitly designated this as an important direction for future work.

***

**Reviewer 96gJ** highlights the work's strong motivation, novelty, and soundness.

> **Main concerns**
* Comparison with DiffStitch.
* Next-state regularization vs. action-level regularization.
* Trade-offs of the T-symmetry prior and evaluation on cloned/expert tasks.

> **Addressed by**
* We provided a detailed comparative analysis with DiffStitch and presented the evaluation results in our response.
* In our responses, we listed evidence showing the drawbacks of using BC-like terms for policy training (comparing TELS against TSRL, TD3+BC within the learned latent space, and TSRL+POR).
* We discussed the trade-offs of the T-symmetry prior in Appendix B.5. Additionally, we added experimental results of TELS on Adroit-cloned and Adroit-expert tasks in Table 6, Appendix B.2 as suggested by the reviewer, where TELS continues to achieve strong performance.

***

In conclusion, we believe our revisions and detailed rebuttals have resolved the primary concerns highlighted during the review phase.

---

### Meta-Review · Area_Chair_2zNU · 2025-12-31

**Summary:**

This paper introduces a T-symmetry enforced inverse dynamics model (TS-IDM) that can learn well-behaved and OOD generalizable latent representations, and facilitate action inference.  The proposed model design is centered around regulating and enforcing its consistency properties, which helps to ensure  a well-behaved latent space can be  learned.

**Reviewer Concerns:**

The reviewers raised a few concerns, including 1) Real-world non-invertible dynamics may violate T-symmetry assumptions, and latent interpretability is limited; 2) Baselines are relatively outdated.  The authors clarified that this paper adopted an extended version of T-symmetry for generic discrete-time MDP settings, namely T-symmetry in the latent state space rather than the raw observation space. As a result, it would  effectively handle non-invertible dynamics and noise. The authors also added some new results on baselines.

**Reviewer Scores:**

The review scores are 6/4/6/8.

---

### Decision · Program_Chairs · 2026-01-26

Accept (Poster)